



# Atlantic Water advection vs glacier dynamics in northern Spitsbergen since early deglaciation

Martin Bartels[1], Jürgen Titschack[1,2], Kirsten Fahl[3], Rüdiger Stein[3], Marit-Solveig Seidenkrantz[4], Claude Hillaire-Marcel[5], Dierk Hebbeln[1]

[1]MARUM – Center for Marine Environmental Sciences, University of Bremen, 28359 Bremen, Germany
[2]SaM – Senckenberg am Meer, Abteilung Meeresforschung, 26382 Wilhelmshaven, Germany
[3]Alfred Wegener Institute for Polar and Marine Research, 27568 Bremerhaven, Germany
[4]Centre for Past Climate Studies and Arctic Research Centre, Department of Geoscience, Aarhus University, 8000 Aarhus C, Denmark
[5]GEOTOP – Université du Québec à Montréal, Montreal, H3C 3P8, Canada

*Correspondence to:* Martin Bartels (mbartels@marum.de)





### Abstract

Atlantic Water (AW) advection plays an important role for climatic, oceanographic and environmental conditions in the eastern Arctic. Situated along the only deep connection between the Atlantic and the Arctic Ocean, the Svalbard Archipelago is an ideal location to reconstruct the past AW advection history and document its linkage with local glacier dynamics, as illustrated in the present study of a sedimentary record from Woodfjorden (northern Spitsbergen) spanning the last ~15500 years. Sedimentological, micropalaeontological and geochemical analyses were used to reconstruct changes in marine environmental conditions, sea-ice cover and glacier activity. Data illustrate a partial breakup of the Svalbard–Barents–Sea Ice Sheet from Heinrich Stadial 1 onwards (until ~14.6 ka BP). During the Bølling-Allerød (~14.6–12.7 ka BP), AW penetrated as a bottom water mass into the fjord system and contributed significantly to the destabilisation of local glaciers. During the Younger Dryas (~12.7–11.7 ka BP), it intruded into intermediate waters while evidence for a glacier advance is lacking. A short-term deepening of the halocline occurred at the very end of this interval. During the early Holocene (~11.7–7.8 ka BP), mild conditions led to glacier retreat, a reduced sea-ice cover and increasing sea surface temperatures, with a brief interruption during the Preboreal Oscillation (~11.1–10.8 ka BP). During the late Holocene (~1.8–0.4 ka BP), a slightly reduced AW inflow and lower sea surface temperatures compared to the early Holocene are reconstructed. Glaciers, which previously retreated to the shallower inner parts of the Woodfjorden system, likely advanced during the late Holocene. In particular, as topographic control in concert with the reduced summer insolation partly decoupled glacier dynamics from AW advection during this recent interval.





## 1 Introduction

In the context of the ongoing global warming, the Arctic is gaining increasing attention from the scientific
community and the general public (e.g., IPCC, 2014; Masson-Delmotte et al., 2013). This is primarily motivated
by extreme events like the 2012 sea ice minimum, the declining summer sea ice cover of recent decades (Stroeve
et al., 2012) and the significant contribution of melting Arctic glaciers to the rising global sea level (e.g., Alley et
al., 2005; Gregory and Huybrechts, 2006; IPCC, 2014; Overpeck et al., 2006). The Arctic is extremely sensitive
to climatic changes and due to various feedback mechanisms, it is warming twice as fast as the global mean – a
phenomenon commonly called the Arctic amplification (Serreze and Barry, 2011). These drastic changes in the
Arctic also likely have far-reaching impacts. Several studies draw a connection to climate phenomena (especially
severe winters) at lower latitudes (e.g., Cohen, 2016; Cohen et al., 2014; Kug et al., 2015; Mori et al., 2014;
Overland et al., 2015; Overland, 2015), although, causal linkages between these concurrent events are open to
discussion (cf., McCusker et al., 2016; Meleshko et al., 2016; Overland et al., 2016).

While various studies suggest that these recent climate changes are induced by human activities (e.g., Abram et
al., 2016; IPCC, 2014), it still remains unclear to which extent natural mechanisms may be involved. Instrumental
climate data rarely cover more than the last century; sea ice observations by satellites did not even start before late
1978 (Stroeve et al., 2012). Hence, data reaching far beyond these instrumental time series are needed to assess
the full range of natural environmental variability, especially the linkage between oceanic current changes and
glacier instability. Sedimentary records covering centuries and millennia offer such archives of natural climate
variations during pre-industrial times, i.e., without any significant anthropogenic impact. Also in the geological
past, the Arctic has experienced periods of warming when ice shelves broke up and glaciers melted dramatically.
The transition from the Pleistocene to the early Holocene, and especially the Holocene Thermal Maximum (HTM),
when enhanced northward advection of Atlantic Water (AW; e.g., Sarnthein et al., 2003) and maximum insolation
were recorded at high northern latitudes (Berger and Loutre, 1991), may help demonstrating ocean–glacier
interactions under a fast warming trend, possibly comparable to the ongoing global warming.

Today, the advection of warm ocean currents is a main contributor to basal glacier melting in both hemispheres,
which eventually leads to global sea level rise (e.g., Cook et al., 2016; Luckman et al., 2015; Straneo et al., 2010).
Being situated at the only deep-sea gateway that connects the Arctic Ocean with the North Atlantic, the Svalbard
Archipelago is an ideal location to investigate changes in the inflow of relative warm AW into the Arctic Ocean
and its influence on the sea ice regime, regional glacier activity and climatic conditions. This passage – the Fram
Strait – is a major gateway for heat advection to high northern latitudes as well as for sea ice export from the Arctic
towards lower latitudes. The majority of Arctic sea ice (~90 %; Rudels, 2009) and about 50% of its total fresh
water (Serreze et al., 2006) are exported through Fram Strait. Due to circulation of the northernmost branch of the
North Atlantic Current along its western slope, Svalbard is bathed by warmer waters than any other Arctic location
(Fig. 1a, b), providing the area with a relatively mild climate. In the present study, we chose a site at the mouth of
Woodfjorden (Fig. 1c), located at the northern margin of Svalbard where the main branch of AW enters the Arctic
Ocean, for the documentation of ocean–ice interactions from deglacial times to the present. In this fjord system,
which is today fed by tidewater glaciers, enhanced sedimentation rates offer a high temporal resolution.

During the Late Glacial Maximum (~21–18 ka BP; Svendsen et al., 2004), Svalbard´s glaciers and icecaps were
part of the Svalbard–Barents–Sea Ice Sheet. This grounded ice sheet covered the entire Barents Sea and expanded
as far as the shelf break of Svalbard (Landvik et al., 1998). It must have been connected to the Fennoscandian Ice





Sheet during the Last Glacial Maximum (Hughes et al., 2016; Landvik et al., 1998; Svendsen et al., 2004). Woodfjorden and its trough constituted one of the main northern gateways for fast flowing ice streams that drained the north-western sector of these ice sheets during the Last Glacial Maximum (Hormes et al., 2013; Ottesen et al., 2007). Whereas the Holocene and in part the late Weichselian have been relatively well documented off (south-)

western Svalbard (e.g., Hald et al., 2004; Jernas et al., 2013; Majewski et al., 2009; Rasmussen et al., 2007, 2012, 2014b; Sarnthein et al., 2003; Skirbekk et al., 2010; Ślubowska-Woldengen et al., 2007; Werner et al., 2011), palaeoceanographic records from northern Svalbard (Chauhan et al., 2014, 2015; Koç et al., 2002; Kubischta et al., 2011; Ślubowska et al., 2005) are still scarce (Fig. 1b) and in most cases based on only few proxies, e.g., stable isotopes, foraminiferal faunal assemblages and/or the analysis of ice-rafted material. The present study aims at

filling this gap, by compiling multi-proxy data, in particular, physical properties of the sediment, the faunal distribution and geochemical composition of benthic foraminifera, as well as biomarkers. This set of proxies offers the opportunity to reconstruct the influence of AW in surface, intermediate and bottom waters, environmental conditions in surface waters and at the bottom water–sediment interface (including sea-ice coverage, primary and export productivity) as well as glacier activity. Additionally, the study site is located at the transition zone from a

glacier-influenced fjord milieu to an open ocean setting; therefore, very sensitive glacier responses to minor changes in oceanic forcing are expected.

## 2 Regional setting

AW is transported towards the Arctic Ocean by the Western Spitsbergen Current (WSC; Fig. 1b). This current follows the western shelf break of the Barents Sea, continuing northward into the eastern Fram Strait (Rudels,

2009) where it sinks and flows below a colder and fresher surface water mass around 78° N (Johannessen, 1986; Manley, 1995). Therefore, heat loss to the atmosphere decreases as the warm AW core deepens (Beszczynska-Möller et al., 2011; Walczowski, 2013). Subsequently, the current is divided when a branch recirculates westward into the Fram Strait between 78–80° N (e.g., Hattermann et al., 2016; Schauer et al., 2004; Fig. 1b), merging with the cold and relatively fresh East Greenland Current that flows southward along the continental shelf of East

Greenland (Rudels, 2009). At the north-western shelf break of Spitsbergen, the WSC is separated into two branches that enter the Arctic Ocean (Fig. 1b). One branch circuits around the Yermak Plateau along its western slope flowing eastward into the Arctic Ocean (Yermak Branch). The second branch, namely the Svalbard Branch, crosses the southern Yermak Plateau and follows the northern shelf of Svalbard (Aagaard et al., 1987). As the Svalbard Branch transports warmer water than the Yermak Branch (Schauer et al., 2004), it is considered as the

most important route for warm, saline AW into the Arctic Ocean (Manley, 1995; Saloranta and Haugan, 2001). These Atlantic-derived waters are divided from colder and fresher Arctic waters by the Arctic (Coastal) Front on the northern shelf of Svalbard (Walczowski, 2013). In the Woodfjorden, inflowing Atlantic-derived waters are stratified during summer – framed by a fresher surface water layer and a deeper colder water mass resulting from fall to winter vertical mixing processes (cf., Cottier et al., 2010), as illustrated by hydrographic data obtained

during cruise MSM02/03 with RV *Maria S. Merian*, in August 2006 (Fig. A1 in the Appendix).

The Woodfjorden system in northern Spitsbergen, the main island of the Svalbard Archipelago, includes the eponymous main fjord and the two tributary fjords Bockfjorden and Liefdefjorden (Fig. 1c). At present, only the latter is fed by four tidewater glaciers: Monacobreen and Seligerbreen (-*breen* = glacier [norw.]), which form one joint glacier front, and the two smaller glaciers Emmabreen and Idabreen (Fig. 1c). Monacobreen is the tenth



largest outlet glacier on Svalbard (Hagen et al., 1993). Meltwater from these glaciers contributes to the cold, relatively low salinity surface water layer in the fjord (Fig. A1). Today, almost 60 % of Svalbard´s landmasses are glaciated (Hagen et al., 1993; Kohler et al., 2007). These almost 34000 km² glaciers and icecaps represent 4.6 % of all glaciated areas on Earth (Radić et al., 2014). Calving of Svalbard´s tidewater glaciers causes a total mass

loss of 5.0–8.4 km³ yr⁻¹ contributing ~2 % (0.02 mm yr⁻¹) to the annual global sea level rise (Błaszczyk et al., 2009).

## 3     Material and Methods

The 275 cm-long sediment core GeoB10817-4 used in this study was retrieved at the mouth of Woodfjorden (N-Spitsbergen; 79.80° N, 14.20° E; water depth 171 m; Fig. 1c) during cruise MSM02/03 in 2006 (Lherminier et al.,

2009). Working halves were sampled with 10 ml syringes for microfossil analyses and grain size measurements at 5 cm intervals, as well as for biomarker analyses in a lower resolution at selected depths.

### 3.1     Computer tomography

Archive halves of the sediment core GeoB10817-4 were scanned by a Toshiba Aquilion 64™ computer tomograph (CT) at the hospital *Klinikum Bremen-Mitte*, with an x-ray source voltage of 120 kV and a current of 600 mA. The

CT image stacks have a resolution of 0.35 mm in x- and y-direction and 0.5 mm resolution in z-direction (0.3 mm reconstruction unit). Images were reconstructed using Toshiba's patented helical cone beam reconstruction technique. The obtained CT data were processed using the ZIB edition of the Amira software (version 2016.25; Stalling et al., 2005). Within Amira, the CT scans of the core sections were merged and core liners, including about 2 mm of the core rims, were removed from the dataset. All clasts > ~1 mm and bioturbation traces were

quantified in each reconstruction slice with the *Segmentation Editor* (threshold segmentation) and the *Material Statistics* module. Used threshold values were: >1500 for lithic clasts, 601–1499 for matrix sediment, 1–600 for bioturbation traces (open voids within the sediment that were filled by air and/or water were included), <1 for the surrounding air and water. With the *Connected Components* module, the individual clasts were separated and subsequently analysed with the *Shape Analysis* module for their parameterisation. The determined clast length was

further used for a clast-size analysis. Therefore, every clast within an interval of 167 CT-slices (corresponds to a ~5 cm core interval) was considered and the obtained result was written to the central slice position. The analysing interval was moved slice by slice. The final results (unit: vol. % of all segmented clasts) were exported in a spreadsheet. Additional the x-ray density of the matrix sediment was evaluated by calculating the mean value and its standard deviation of the matrix sediment per slice (measured in Hounsfield units (HU); the matrix sediment

segmentation was reduced by two voxels to exclude potential marginal artefacts).

### 3.2     AMS Radiocarbon measurements

The chronostratigraphy of core GeoB10817-4 is based on 11 AMS radiocarbon measurements from either mixed benthic foraminifera or *N. labradorica* tests, achieved at the *CologneAMS*, facility of the University of Cologne, or at the ETH of Zürich, respectively (Table 1). Radiocarbon ages were converted into calibrated years BP

(= before 1950 C.E.) using the radiocarbon calibration software CALIB 7.1 (calib.qub.ac.uk/calib/) and the Marine13 calibration curve (Reimer, 2013). Because the exact local reservoir age at the core site – which might have varied over time – is unknown, an averaged regional reservoir age correction (ΔR = 98±37 years) was used,



based on six ΔR values from the Svalbard region (Mangerud, 1972; Mangerud and Gulliksen, 1975; Olsson, 1980) derived from the Marine Reservoir Correction Database (calib.qub.ac.uk/marine/).

### 3.3 Grain size measurements

Grain size measurements were performed in the Particle-Size Laboratory at MARUM, University of Bremen with a Beckman Coulter Laser Diffraction Particle Size Analyser LS 13320™. Prior to these measurements, the terrigenous sediment fractions were isolated by removing organic carbon, calcium carbonate, and biogenic opal by boiling the samples (in about 200 ml water) with 10 ml of $H_2O_2$ (35 %; until the reaction stopped), 10 ml of HCl (10 %; 1 min) and 6 g NaOH pellets (10 min), respectively. After each step the samples were diluted (dilution factor: >25). Finally, remaining aggregates were destroyed prior to the measurements by boiling the samples with ~0.3 g tetra-sodium diphosphate decahydrate ($Na_4P_2O_7 * 10H_2O$; 3 min; cf., McGregor et al., 2009). Sample preparation and measurements were carried out with deionized, degassed and filtered water (filter mesh size: 0.2 μm) to reduce the potential influence of gas bubbles or particles within the water. Results consist in grain-size distributions within a 0.04 to 2000 μm range, divided into 116 size classes. The calculation of the grain sizes relies on the Fraunhofer diffraction theory and the Polarization Intensity Differential Scattering (PIDS) for particles from 0.4 to 2000 μm and from 0.04 to 0.4 μm, respectively. Reproducibility is checked regularly through replicate analyses of three internal glass-bead standards. It is better than ±0.7 μm for the mean and ±0.6 μm for the median grain size (1σ). The average standard deviation integrated over all size classes is better than ±4 vol. % (note that the standard deviation of the individual size classes is not distributed uniformly). All provided statistic values are based on geometric statistics.

Sediment grains larger than 310 μm have not been detected by the laser diffraction particle size analyser although coarser sand particles and even gravels are detected within the sieved samples of the microfossil analysis as well as within the CT clast-size analyses (see sect. 3.4 and sect. 4.1; Fig. 2a). This bias results from the very small sample volume (~0.5 ml) for the particle size analysis. However, taking into account that >98 wt. % of the dried bulk sediment is <150 μm (with the exception of 4 samples with 3, 4, 6 and 12 wt. % >150 μm fraction; Bartels, unpublished data), the observed grain size patterns of the siliciclastic fraction are considered as reliable.

### 3.4 Microfossil analyses

Bulk sediment samples were washed through a 63 μm sieve. The coarse fraction was dried and sieved through 100 μm and 150 μm meshes. Faunal analyses were carried out on two fractions: 100–150 μm and >150 μm (Fig. A2 in the appendix). Counts from both fractions were summed up (i.e., >100 μm) for the discussion below. Samples with high numbers of foraminifera were split with a micro-splitter. Wherever possible, 200–300 benthic foraminifera were counted in each fraction of every sample to obtain statistically robust results (cf., Murray, 2006; Patterson and Fishbein, 1989). In addition, three replicates of a non-split as well as of a split sample were counted (each split was counted four times, including the initially counted sample). Statistical calculations reveal a standard error of 3.52 and 2.95 %, respectively. Most benthic foraminifera were identified at species level except for agglutinants that were mainly identified at genera level due to their poorer preservation. The majority of calcareous benthic foraminifera are well preserved, but several specimens show indications of dissolution or transport. Small fragments were not counted since these specimens might be allochthonous. Distinct *Stainforthia-* and *Buccella-* morphotypes were lumped together as *Stainforthia loeblichi* s.l. and *Buccella frigida* s.l., respectively, because in



each case possibly two species (*S. loeblichi* s.s. and *S. concava* as well as *B. frigida* s.s. and *B. tenerrima*, respectively) could not be distinguished definitely. Hyaline and agglutinating species that could not be identified are summarized as Rotaliina and Textulariina, respectively. Only few planktonic foraminifera were present in the material. As post-mortem degradation might be the main reason for the distribution of arenaceous taxa (Murray, 2006; cf., Hald and Korsun, 1997; Ślubowska et al., 2005), percentages of calcareous benthic species were calculated excluding agglutinants to avoid these taphonomic effects.

Accumulation rates of benthic foraminifera were calculated following the equation of Ehrmann & Thiede (1985):

$$\mathrm{ARBF} = \mathrm{LSR} * \rho * \mathrm{BFN} \ (1)$$

with:

ARBF = accumulation rate of benthic foraminifera [individuals $cm^{-2}$ $ka^{-1}$];

LSR = linear sedimentation rate [cm $ka^{-1}$];

$\rho$ = density of dry sediment [g $cm^{-3}$];

BFN = Benthic Foraminifera Number (per gram dry sediment) [individuals $g^{-1}$]

### 3.5 Stable isotope measurements

Stable carbon and oxygen isotopes were measured at ZMT (Leibniz Centre for Tropical Marine Ecology) in Bremen, using a Finnigan MAT 253™ gas isotope ratio mass spectrometer with a Kiel IV automated carbonate preparation device (standard deviations of house standard: 0.02 ‰ for $\delta^{13}$C and 0.06 ‰ for $\delta^{18}$O), and at MARUM (Bremen) using a Finnigan MAT 251™ mass spectrometer with a Kiel I carbonate preparation device (standard deviations of house standard: 0.02 ‰ for $\delta^{13}$C and 0.03 ‰ for $\delta^{18}$O). The measurements were performed on 2–8 tests (test size: 150–500 μm) of the epibenthic foraminifera *Cibicides lobatulus* and the endobenthic foraminifera *Nonionellina labradorica* for bottom water and pore water signals, respectively. Stable oxygen isotope values were corrected for a global ice effect (Waelbroeck et al., 2002) and adjusted for vital effects: +0.64 ‰ for *C. lobatulus* (Shackleton, 1974), -0.2 ‰ for *N. labradorica*: (Duplessy et al., 2005).

### 3.6 Biomarker analyses

Selected dried sediment samples were homogenised and used for geochemical analyses. 100 μg of sediment were used for the determination of the total organic carbon (TOC) content using an ELTRA™ Analyser. Alkenone analyses, carried out to reconstruct sea surface temperatures (SST), were extracted using 2 g of sediment with an Accelerated Solvent Extractor (DIONEX™, ASE 200; 100 °C, 1000 psi, 15 min, dichloromethane and methanol (99:1, v/v) as solvent). The separation of compounds was carried out by open column chromatography ($SiO_2$) using *n*-hexane and dichloromethane (1:1, v/v), and dichloromethane. The composition of alkenones was analysed by gas chromatography (Agilent™ 7890). Individual alkenone ($C_{37:4}$, $C_{37:3}$, $C_{37:2}$) identification is based on retention time and the comparison with an external standard.

The alkenone unsaturation index ($U^K_{37}$) as proxy for SST (°C) was calculated following Brassell et al. (1986) due to the presence of the $C_{37:4}$ alkenone. For calculating SST, the following equation

$$U^{K'}_{37} = 0.033 * T \ (°C) + 0.044 \ (2)$$



by Müller et al. (1998) – based on a global core-top calibration (60° N–60° S) – was used (replacing $U_{37}^{K'}$ by $U_{37}^{K}$ due to the presence of the $C_{37:4}$ alkenones). This equation (Eq. 2) yielded more realistic results compared to other whose temperature reconstructions were much to high (e.g., Prahl and Wakeham, 1987). The standard error of the calibration is reported as ±1.5 °C. The instrument stability was continuously controlled by re-runs of an external

alkenone standard (extracted from *Emiliania huxleyi* cultures with known growth temperature) during the analytical sequences. The range of the total analytical error calculated by replicate analyses is less than 0.4 °C.

The $C_{25}$ isoprenoid lipid biomarker ($IP_{25}$) was analysed to document sea ice coverage (cf., Belt et al., 2007). Brassicasterol and dinosterol have been used as phytoplankton biomarkers (cf., Fahl and Stein, 1999; Volkman, 1986; Volkman et al., 1993). For extraction of $IP_{25}$ and sterols, 3 g of sediment were ultrasonicated (Sonorex Super

RK 510, 35 Khz, 15 min) using dichloromethane:methanol (2:1, v/v) as solvent. For quantification, internal standards 7-hexylnonadecane (7-HND, 0.076 µg per sample for $IP_{25}$ quantification), squalane (2.4 µg per sample) and cholest-5-en-3β-ol-$D_6$ (10 µg per sample for sterol quantification), were added prior to analytical treatment. Separation of the hydrocarbon and sterol fractions was carried out via open column chromatography (hydrocarbon fraction with 5 ml *n*-hexane, the sterol fraction with 6 ml *n*-hexane:ethylacetate (5:1, v/v)). The latter fraction was

silylated with 500 ml BSTFA (bis-trimethylsilyl-trifluoroacet-amide) (60 °C, 2h). $IP_{25}$ and sterols were analysed by gas chromatography/mass spectrometry. Component assignment was based on comparison of gas chromatography (GC) retention times with those of reference compounds and published mass spectra. The Kovats Index calculated for $IP_{25}$ is 2086. The detection limit for quantification of $IP_{25}$ (Agilent 7890B GC™, Agilent 5977A™ Extractor MSD with Performance Turbo Pump) is 0.005 ng µL$^{-1}$ in SIM (selected ion monitoring) mode.

The retention indices for brassicasterol (as 24-methylcholesta-5,22E-dien-3β-O-Si(CH₃)₃) and dinosterol (as 4α,23,24-trimethyl-5α-cholest-22E-en-3β-O-Si(CH₃)₃) were calculated to be 1.018 and 1.091 (normalized to cholest-5-en-3β-ol-$D_6$ set to be 1.000), respectively.

For the quantification of $IP_{25}$, its molecular ion (*m/z* 350) in relation to the abundant fragment ion *m/z* 266 of the internal standard (7-HND) was used (SIM mode). The different responses of these ions were balanced by an

external calibration (Fahl and Stein, 2012). Brassicasterol and dinosterol were quantified as trimethylsilyl ethers using the molecular ions *m/z* 470 and *m/z* 500, respectively, in relation to the molecular ion *m/z* 464 of cholesterol-$D_6$. For more details about the quantification of $IP_{25}$ as well as the sterols see Fahl and Stein (2012). Accumulation rates of sea ice and phytoplankton biomarkers ($IP_{25}$ and brassicasterol, respectively) were calculated following Eq. (1) by replacing the BFN term accordingly.

For more semi-quantitative estimates of present and past sea ice coverage, Müller et al. (2011) combined the sea ice proxy $IP_{25}$ and phytoplankton biomarkers in a phytoplankton-$IP_{25}$ index, the so-called "$PIP_{25}$ index":

$$PIP_{25} = \frac{[IP_{25}]}{[IP_{25}] + ([phytoplankton\ marker] * c)} \quad (3)$$

with c = mean $IP_{25}$ concentration/mean phytoplankton biomarker concentration.

As phytoplankton biomarker, brassicasterol was used (for further discussion of advantages and limitations of the

$PIP_{25}$ approach see Belt and Müller, 2013; Smik et al., 2016; Stein et al., 2012; Xiao et al., 2015). Following the classification scheme of Müller et al. (2011), $PIP_{25}$ values between 0.3 and 0.5, 0.5 and 0.7, and >>0.7 point to a reduced sea ice cover, a seasonal sea ice cover including an ice edge situation, and an extended to perennial sea ice cover, respectively.





## 4    Results

### 4.1    Core description

The sediment of core GeoB10817-4 consists of some coarse sand and gravel, with scattered pebbles (Fig. 2a) embedded in a homogeneous silty clay matrix sediment. The upper ~30 cm of the sediment core are heavily

bioturbated, as illustrated by the CT scan. The interval between 31 and 54 cm is characterised by a basal erosional unconformity, weakly visible on the CT picture (Fig. 2b), and high amounts of (extra- and intra-) clasts (Fig. 2b), sand (including high quantities of foraminiferal tests), and bivalve shell fragments. Furthermore, the interval exhibits an increasing x-ray density with an enhanced standard deviation accompanied by a coarsening upward of the siliciclastic fine fraction (<63 µm; observed in the grain-size analyses; Fig. A3 in the appendix). These

sedimentological features point to an allochthonous origin and a deposition by mass-wasting. Consequently, this interval is excluded from the subsequent palaeoceanographic reconstructions. Below this disturbance and as deep as 140 cm, a small amount of clasts is found, whereas the lower part of the core contains high quantities of coarse debris (Fig. 2a).

### 4.2    Chronology

Assuming linear sedimentation rates between dated depths (ranging between 8 and 121 cm ka$^{-1}$), calibrated ages have been interpolated (Fig. 2c; Table 1). The segment below the 31-54 cm interval (sect. 4.1) covers a time span from ~15.7–7.8 cal. ka BP whereas the above one ranges from ~1.8–0.3 cal. ka BP (from here on, all ages are given as cal. ages BP and data are presented exclusively against age). This indicates that the deposition of the 31–54 cm layer led to a ~6000 years gap in the sequence, spanning a large part of the Holocene. The radiocarbon age

at 257 cm depth has been excluded from the age model because it was obtained from a much smaller sample (~0.4 mg carbonate) than the overlying 237 cm-deep older sample (~4.3 mg) that has been measured four times with consistent results.

### 4.3    Sedimentology

#### 4.3.1    Grain sizes

Until ~13 ka BP, the grain-size distribution of siliciclastic sediments reached a maximum of about 7 µm (mean: 4–6 µm). In the subsequent period until ~11.1 ka BP, a slight coarsening (max. at ~8–9 µm, mean: 6–7 µm) is observed. A further shift to coarser silt (max. at ~20 µm, mean: 8–9 µm) appeared around 10.8 ka BP. From ~9.4 ka BP on, silt around 45–50 µm shows highest percentages (mean: 11–15 µm). In the uppermost interval of the core (~1.6–0.4 ka BP), a trend back to finer sediments (mean: 7–10 µm) is evident, but with a more even

distribution than below the disturbed section (Fig. 3a, b).

#### 4.3.2    Ice-Rafted Debris

Clasts > ~1 mm detected by CT analyses (sect. 3.1) were considered as ice-rafted debris (IRD). The interpretation of the IRD abundances is based on the volume percentages and numbers of clasts observed on CT scans, as the combination of these data depicts the best available IRD distribution (Fig. 2a, Fig. 3c): the single consideration of

volume percentages would overestimate large clasts while the numbers of clasts alone would overestimate the significance of small debris. CT data show that IRD are more abundant at core bottom, containing high quantities



prior to ~14.5 ka BP (up to 10 vol. %; up to 2.3 clasts cm$^{-3}$), but decrease afterwards (down to 0 % and 0.4 clasts cm$^{-3}$). From ~13.1 ka BP on, IRD amounts rise again until ~11.5 cal. ka BP (max. 7 %; max. 2.8 clasts cm$^{-3}$). From then on, the volume percentages and numbers of clasts illustrate a low but continuous flux of ice-rafted material (mean: 0.2 %; 0.3 clasts cm$^{-3}$), and some increase particularly from ~8.5 ka BP on (up to 1.9 % and 0.8 clasts

cm$^{-3}$). Above the reworked layer (from ~1.8 ka BP on), a further increase of the IRD content is identified especially for larger clasts (up to 7 % and 0.8 clasts cm$^{-3}$; Fig. 3c). Some peaks in volume percent reflect single large clasts (marked with a star in Fig. 3c).

### 4.4    Microfossil analyses

### 4.4.1    Faunal composition

Planktic foraminifera were rare or absent. Their percentages range between 0 and 2 % of the total fauna apart from one sample dated ~13 ka BP (~5 %). Thus, only benthic species were taken into account for calculations and interpretation. In total, 83 taxa (61 calcareous species) were identified (7–56 taxa/sample, 7–39 calcareous taxa/sample; Table A1 in the appendix). Calcareous species represent the major part of the benthic fauna (~80–100 %), agglutinating species occur mainly in the upper part above the reworked layer (up to ~20 %) with

decreasing percentages down-core (mean: ~2 %).

### 4.4.2    Relative abundance and accumulation rate (flux) of benthic foraminifera

The calcareous fauna is dominated by three species: *Cassidulina reniforme* (~18–78 %, Fig. 4g), *Nonionellina labradorica* (~1–60 %, Fig. 4d) and *Elphidium clavatum* (formerly named *E. excavatum* forma *clavata*, see Darling et al., 2016) (~2–46 %, Fig. 4f), accompanied by lower percentages of *Buccella frigida* s.l. (max. ~9 %, Fig. 4b),

*Cassidulina neoteretis* (max. ~23 %, Fig. 4e), *Cibicides lobatulus* (max. ~5 %, Fig. 4a), *Islandiella helenae* (max. ~10 %, Fig. 4c), *Islandiella norcrossi* (max. ~8 %, not shown) and *Stainforthia loeblichi* s.l. (max. ~8 %, not shown).

Until ~12.5 ka BP the accumulation rate of benthic foraminifera is relatively low (mean value: ~2250 individuals (ind.) cm$^{-2}$ ka$^{-1}$) subsequently slightly increasing up to a mean value of ~4770 ind. cm$^{-2}$ ka$^{-1}$. Apart from a few

exceptions, the benthic foraminifera flux is further rising from ~11.2 ka BP on (mean value: ~12980 ind. cm$^{-2}$ ka$^{-1}$) with peaks at ~10.8, ~10.6 and ~10.1 ka BP exceeding 30000 ind. cm$^{-2}$ ka$^{-1}$ (Fig. 6d). Above the disturbed section (from ~1.6 ka BP on), the mean benthic foraminifera flux reaches ~6530 ind. cm$^{-2}$ ka$^{-1}$. Apart from *C. neoteretis* (which shows peak fluxes exceeding 500 ind. cm$^{-2}$ ka$^{-1}$ at ~13 and ~11.9 ka BP; Fig. 4e), flux rates of each (dominant) species generally follow the pattern of the total benthic foraminiferal flux (Fig. 4).

### 4.5    Geochemistry

### 4.5.1    Stable isotopes

Stable oxygen isotope values, corrected for vital effects and the global ice effect (sect. 3.5), increase slightly towards core top with *C. lobatulus* showing more variability and lighter values than *N. labradorica* (ranging from 2.5 to 4.1 ‰ and from 3.6 to 4.4 ‰, respectively). Between ~12 and ~9 ka BP, δ$^{18}$O of both species exhibit an

enhanced offset (Fig. 5a). Stable carbon isotope values show minor variations for *C. lobatulus*, but a slight increase towards core top (0.4 to 1.6 ‰). *N. labradorica* exhibits depleted δ$^{13}$C values from ~11 to ~10.1 ka BP (down



to -2.7 ‰; Fig. 5b). The $\delta^{13}$C-offset between epi- and endobenthic species ($\Delta\delta^{13}$C) varies between 1.7 and 3.7 ‰, with peak values at ~12.6, ~10.6, ~10.4 ka BP and ~0.4 ka BP (Fig. 5c).

### 4.5.2 Biomarkers

Alkenone unsaturation index ($U^{K}_{37}$) values range from 0.1 to 0.3. They yield SST values ranging from ~1 to ~7 °C. Highest temperatures were attained between ~11 and ~10 ka BP, whereas the lowest values are calculated prior to ~11.9 ka BP (Fig. 6a).

Accumulation rates of the $C_{25}$ isoprenoid lipid biomarker ($IP_{25}$) range between 0.8 µg cm$^{-2}$ ka$^{-1}$ (~11 ka BP) and almost 0 µg cm$^{-2}$ ka$^{-1}$ (~9.4 ka BP) (Fig. 6c). The phytoplankton-$IP_{25}$ index ($PIP_{25}$ index) using the phytoplankton biomarker brassicasterol (cf., Müller et al., 2011) exhibits its highest values at ~12.8 ka BP (~0.9) and its lowest values at ~9.4 ka BP (~0.1) (Fig. 6b).

The accumulation rates of brassicasterol range between ~1.5 µg cm$^{-2}$ ka$^{-1}$ (~14.2 ka BP) and ~167.5 µg cm$^{-2}$ ka$^{-1}$ (~10.9 ka BP) (Fig. 6d). The accumulation rate of dinosterol shows an almost parallel trend to that of brassicasterol but values are one magnitude lower (Fig. A4 in the appendix).

## 5 Interpretation and discussion

### 5.1 Late Weichselian (until ~12.7 ka BP) – deglaciation in the Woodfjorden area

In the Woodfjorden area, the Heinrich Stadial 1 (until ~14.6 ka BP) was characterised by a very high IRD content (Fig. 7b) signifying enhanced calving of the glacier front. The increased percentages of fine siliciclastic sediments within this interval (mean: ~4–5 µm; Fig. 7a) point to some contemporaneous intensified meltwater outflows (in this study, sources of siliciclastic sediments were interpreted as terrigenous). Thus, it is assumed that parts of the Svalbard–Barents–Sea Ice Sheet disintegrated during Heinrich Stadial 1 while stable oxygen isotope records of the NGRIP ice-core show that atmospheric temperatures remained cold (Fig. 7g; Rasmussen et al., 2014a). Contrastingly, bottom waters might have been relative warm and/or saline as indicated by the appearance of the benthic foraminiferal species *C. neoteretis* (Fig. 7d) as well as the high percentages (up to ~80 %) of *C. reniforme* (Fig. 4g). The occurrence of *C. neoteretis* has been connected with a strong influence of AW in bottom waters because this species is assumed to respond to a limited temperature and salinity range as well as to an enhanced food supply (Jennings et al., 2004; Rytter et al., 2002; Seidenkrantz, 1995; Seidenkrantz et al., 2013; Steinsund, 1994). *C. reniforme* also prefers saline bottom waters (Hald and Korsun, 1997; Jennings et al., 2004) and may therefore be linked to the intrusion of AW into bottom waters. The advection of AW as a subsurface or even bottom water mass during Heinrich Stadial 1 as well as during earlier Heinrich stadials, has already been proposed for the Svalbard area by Rasmussen et al. (2007, 2014c). Nowadays, basal melting of tidewater glaciers in Spitsbergen by high advection rates of the relatively warm AW, is evident (Luckman et al., 2015). Thus, during Heinrich Stadial 1, the influx of AW in the Woodfjorden area possibly contributed to the destabilisation of glacier fronts. Apart from oceanic heat advection, topographic features of the relative wide Woodfjorden system may have promoted the glacier retreat. Gump et al. (2017) reported about a more than thousand years delayed deglaciation of narrower fjords with higher surrounding mountains in south-western Norway (see also Stokes et al., 2014).

During the Bølling (~14.6–14 ka BP), a continuing outflow of meltwater is expected as the grain-size distribution still exhibits a peak around 7 µm (mean: ~4 µm; Fig. 3a, b and Fig. 7a). Synchronously, the sediment shows a





reddish colour (Munsell code 10R4/6) probably connected to sediment plumes loaded by glacial flour from the erosion of Devonian *Old Red* sandstones. Source rocks, i.e., the Wood Bay Formation, crop out at the north-western as well as at the south-eastern coast of Woodfjorden (Reinsdyrflya and Andrée Land, respectively; Fig. 1c) (Harland, 1998). Melting glaciers from the former lowland may have been a potential source of these reddish
sediments. This interpretation is supported by [10]Be ages of erratic boulders which suggest that Reinsdyrflya started to be deglaciated during Heinrich Stadial 1 and became ice-free during the Bølling-Allerød (Gjermundsen et al., 2013; Hormes et al., 2013). Rasmussen and Thomsen (2013) reported on "pink" sediment layers derived from Devonian Red Beds in sediment cores from the north-western slope of Svalbard. They interpreted those layers as deposited by meltwater plumes during interstadials. Forwick et al. (2010) also suggested that meltwater run-off
connected to retreating glaciers was one possible source for fine reddish sediments deposited close to glacier-fronts. Correspondingly, a glacier-proximal position is likely at our core site where tidewater glaciers may have successively waned. Percentages of *C. neoteretis* reached their maximal values during the Bølling (Fig. 7d). Almost synchronously, this taxon peaked at the south-western (Rasmussen et al., 2007) and at the northern margin of Svalbard (Ślubowska-Woldengen et al., 2007; Ślubowska et al., 2005) as well as in the Laptev Sea (off northern
Siberia; Taldenkova et al., 2010) – although with considerable higher values probably deriving from a more direct influence of AW and greater water depths compared to our core location. Massive meltwater outbursts connected to the retreating glaciers possibly caused a deepening of the halocline (cf., Rasmussen and Thomsen, 2004) as the extended cold and relative fresh surface water mass forced the AW to submerge. Accordingly, cold surface waters have been insulated from those relative warm bottom waters enabling an extended sea ice coverage (cf., SST and
sea ice cover; Fig. 7).

In the course of the Allerød (~14–12.7 ka BP), the grain-size distribution shows a general trend to slightly coarser sediment (mean: ~5.5–6 μm; Fig. 7a). The coarsening may reflect the retreat of some glaciers to more distal positions. Consequently, less fine-grained material from meltwater plumes would have reached the core location. Szczucin and Zajączkowski (2012) showed that meltwater-derived suspended particular matter is mostly deposited
within few kilometres from sources in glacier-influenced fjords of Svalbard, as mixing of meltwater with sea water induces flocculation and a subsequent suspension settling. A further retreat of the glaciers is also suggested by slightly increasing amounts of (most likely ice-rafted) clasts in the studied core (Fig. 7b), indicating enhanced calving. Radiocarbon-dated beach ridges at the eastern coast of Woodfjorden (Andrée Land) point to a glacio-isostatic emergence of ~20 m (Forman et al., 2004) in the course of the Allerød (Fig. 7f). This uplift illustrates a
massive retreat of the Svalbard–Barents–Sea Ice Sheet, at least following Heinrich Stadial 1, as the decreasing ice sheet load caused a successive continental rebound.

Towards the transition to the Younger Dryas (YD), percentages as well as accumulation rates of *C. neoteretis* increased again (Fig. 4e and Fig. 7d). This trend is also in agreement with studies from the western, south-western and northern Svalbard margin (Rasmussen et al., 2007; Ślubowska-Woldengen et al., 2007; Ślubowska et al.,
2005), although percentages of *C. neoteretis* were significantly higher at these locations during the Allerød-YD transition. The concomitant appearance of *C. neoteretis* may indicate some AW influx reaching the sea floor (e.g., Lubinski et al., 2001; Steinsund, 1994).



### 5.2 Younger Dryas (~12.7–11.7 ka BP) – oceanographic transitions and glacier responses

During the early YD, the benthic foraminiferal fauna was still dominated by *C. reniforme* (Fig. 4g) but percentages of *N. labradorica* rapidly increased (Fig. 7c). The latter species commonly blooms in connection to enhanced productivity found near oceanic fronts (e.g., Rytter et al., 2002; Sheldon et al., 2016b; Steinsund, 1994).
Ślubowska-Woldengen et al. (2007) concluded that rising percentages of *N. labradorica* reflect the approaching Arctic Coastal Front, which divided Atlantic (i.e., the Svalbard Branch) and Arctic waters. Studies from the northern Svalbard continental slope (Koç et al., 2002; Ślubowska et al., 2005) as well as from the Bellsund trough (western Spitsbergen; Ślubowska-Woldengen et al., 2007) show similar distributions of *N. labradorica* during the YD. At the study site, AW may have risen in the water column, possibly detaching from the seafloor and enabling
the formation of denser Winter Water at the bottom (cf., modern situation illustrated in Fig. A1). The rise of AW may have been caused by a slightly declined meltwater outflow (Fig. 3a, Fig. 7a), which no longer forced AW to submerge, and/or by upwelling events. Modern AW intrusion into the fjords of Spitsbergen occurs intermittently (cf., Carmack and Wassmann, 2006; Wassmann, 2015) and is amplified by wind-induced upwelling (cf., Cottier et al., 2005, 2007). Today, easterlies are the prevailing winds at the northern margin of Svalbard (Lind and
Ingvaldsen, 2012). Then, northward Ekman transport of surface waters generates AW upwelling, which subsequently is able to flood the northern shelfs of Svalbard (Falk-Petersen et al., 2015; Lind and Ingvaldsen, 2012). A northward retreat of the sea-ice edge even amplifies this mechanism (Falk-Petersen et al., 2015 and references therein). Comparable conditions might have occurred very frequently at the mouth of Woodfjorden during the early YD, especially because the sea-ice coverage was temporarily reduced during this time interval
(Fig. 6b). Due to the resultant shallower halocline, an oceanic front (i.e., the Arctic Coastal Front) formed, separating AW from Arctic Water. In the Arctic, AW is commonly associated with enhanced nutrition: Nutrients are transported from deeper water masses to the surface in the North Atlantic by winter mixing, the subsequent northward advection of these nutrients promotes primary production at higher northern latitudes while Arctic-derived waters contain only low nutrient amounts (Hunt et al., 2016). Thus, the advection of AW distributes
nutrients onto the Arctic shelfs, e.g., the shelfs and fjords of Svalbard (Carmack and Wassmann, 2006). Therefore, the inflow of (sub-)surface AW in the photic zone is interpreted as leading directly to a higher productivity (Carmack and Wassmann, 2006; Falk-Petersen et al., 2015; Sakshaug, 1997). Accordingly, nutrient advection and consequential increasing phytoplankton blooms in spring/early summer may have resulted from such upwelling events during early YD. High $\delta^{13}$C-offsets ($\Delta\delta^{13}$C; Fig. 7e) between the epi- (*C. lobatulus*) and the endobenthic
species (*N. labradorica*) during this time interval point to enhanced remineralisation rates and consequential enhanced export production (cf., Mackensen, 2008; Mackensen and Schmiedl, 2016). Apart from a generally amplified productivity resulting from the stronger summer insolation of the YD (Fig. 7g; Berger and Loutre, 1991), as suggested by Müller and Stein (2014), the increased (export) production possibly resulted from AW incursion into the photic zone. The subsurface advection of the relatively warm AW may have also contributed to further
retreat of local glaciers: Maximum amounts of IRD (Fig. 7b) and risen percentages of fine-grained sediments (mean: ~6 µm; Fig. 7a) point to increased calving rates and meltwater outflow.

Following the peak appearance of *N. labradorica*, a shift in the faunal distribution is recognised around 12.4 ka BP: Percentages of *C. reniforme* declined while those of *E. clavatum* rose (Fig. 4d, f, g). The dominance of *E. clavatum* possibly responded to comparable unstable conditions linked to sediment-laden waters and/or high-
frequency IRD-deposition connected to glacier melting events. *E. clavatum* is found in extreme modern





environments, e.g., affected by very turbid meltwater plumes and/or relative low salinities, where *C. reniforme* is not able to survive (Korsun and Hald, 1998; Steinsund, 1994). A shift from relatively stable to unstable conditions during YD has been inferred from various studies, although this shift may have started at the earliest at ~12.3–12.2 ka BP (Bakke et al., 2009; Lane et al., 2013; Pearce et al., 2013). The slightly earlier inception at Woodfjorden

may result from dating uncertainties and/or regional differences (cf., Lane et al., 2013).

Cabedo-Sanz et al. (2013) report about ameliorating, sea-ice-free conditions off northern Norway as early as ~11.9 ka BP. In our record, lower IRD deposition (Fig. 7b) along with gradually declining percentages of *E. clavatum* (Fig. 4f) are observed during the late YD (from ~12 ka BP on), which probably indicates a further retreat of glacier fronts into inner fjord positions with a consequential reduction in icebergs melting at the mouth of Woodfjorden.

Salvigsen and Høgvard (2006) noted that several glaciers at the head of Bockfjorden, a tributary fjord of the inner Woodfjorden (Fig. 1c), have been disconnected from main fjord at the end of the YD. Prevailing fine-grained sediments (mean: ~6 µm; Fig. 7a) – presumably connected to meltwater run-off – also point to some disintegration of glaciers. Birgel and Stein (2004) assumed that a "last deglaciation event" on Svalbard occurred around 12 ka BP, although they proposed a concurrent stable AW advection warming surface waters. Contrastingly, our data

indicate that the spreading of cold and low salinity surface waters – resulting from enhanced inputs of meltwater – likely deepened the Atlantic-derived waters. Accordingly, AW-influenced bottom waters may have resulted in the observed slightly increasing percentages and peaking accumulation rates of *C. neoteretis* (Fig. 4e, Fig. 7d) during late YD (cf., Jennings et al., 2004; Rytter et al., 2002; Steinsund, 1994). As surface waters were consequently better insulated from relative warm AW, SSTs probably fell again while sea-ice expanded (cf., Fig.

6a, b). However, a perennial closed sea-ice cover in Woodfjorden, as suggested by Brückner and Schellmann (2003), seems unlikely as the above data show continuous deposition of IRD during the entire YD (Fig. 7b), even when maximum $PIP_{25}$ values are observed (Fig. 6b), leading to infer broken-up sea-ice at least during summer to enable high productivity of sea-ice diatoms and icebergs calving and dispersal.

The retreat of glaciers during the YD as suggested by our data is further supported by studies based on radiocarbon-

dated beach ridges or moraines (Mangerud and Landvik, 2007; Salvigsen and Høgvard, 2006; Salvigsen and Österholm, 1982) as well as on $^{10}$Be-dated boulders (Reusche et al., 2014). They point to the absence of any YD glacier advance along the western and northern coasts of Spitsbergen. Possible contemporaneous glacier advances in several other regions of Svalbard – especially in Isfjorden (central Spitsbergen) – still remain under debate (e.g., Birgel and Stein, 2004; Boulton, 1979; Forwick, 2005; Svendsen et al., 1996).

**5.3    Early Holocene (~11.7–7.8 ka BP) – amelioration of environmental conditions**

Climatic conditions ameliorated during the YD/early Holocene transition (starting ~11.7 ka BP) as summer insolation approached its maximum values (Fig. 7g; Berger and Loutre, 1991). Also the high offset between $\delta^{18}O$ values of *C. lobatulus* and *N. labradorica* (Fig. 5a) – which already started during the second half of the YD and persists during the entire early Holocene – presumably indicates increasing seasonal differences due to the contrast

between high summer vs low winter insolation (Fig. 7g). Indeed, *C. lobatulus* calcifies in summer, whereas *N. labradorica* calcifies in spring or early summer (Zajączkowski et al., 2010). Rising $\Delta\delta^{13}C$ and depleted $\delta^{13}C$ values in *N. labradorica* (Fig. 5b, Fig. 7e) may indicate higher organic matter inputs deriving from enhanced primary production. High percentages of *N. labradorica* (Fig. 7c) point to a growing influence of the Arctic Coastal Front with intensified AW circulation in intermediate waters (cf., sect. 5.2). Synchronously, declining IRD





amounts suggest a significant reduction in iceberg rafting (Fig. 7b). Grain-size measurements illustrate some coarsening (Fig. 7a), likely resulting from the retreat of glaciers to innermost positions in the fjords as smaller amounts of fine-grained suspended material associated with meltwater plumes reached the core location (cf., Szczucin and Zajączkowski, 2012).

Between ~11.1 and ~10.8 ka BP, a short-term shift in the faunal assemblage is recognised: Percentages of *N. labradorica* dropped while those of *E. clavatum* and *C. reniforme* increased (Fig. 4d, f, g). This likely illustrates a northward shift of the Arctic Coastal Front (away from the coast) and a consequential stronger influence of cold and fresh Arctic waters. Also some surface cooling is indicated by rapidly decreasing SSTs (Fig. 6a). *E. clavatum* – an opportunistic species – benefitted from these deteriorated conditions. Increasing percentages of *C. reniforme*

may reflect the presence of some saline AW at the fjord floor (cf., Jennings et al., 2004). Several studies describe simultaneous and similar faunal changes around Svalbard (Rasmussen et al., 2012, 2014b; Skirbekk et al., 2010; Ślubowska-Woldengen et al., 2007; Ślubowska et al., 2005), generally associated with the *Preboreal Oscillation* (PBO) cooling interval (cf., Björck et al., 1996). Some divergence in the timing of the PBO – especially at Hinlopen trough (Ślubowska et al., 2005) – may be due to chronological uncertainties and possibly distinct reservoir ages,

as suggested by Rasmussen et al. (2014b). Different trigger mechanisms have been suggested for the PBO inception. Meltwater discharge from the disintegrating Fennoscandian Ice Sheet into the Baltic Ice Lake and subsequent drainage into the Nordic Seas have been proposed as one possible cause of the PBO cooling (Björck et al., 1996, 1997; Hald and Hagen, 1998). In contrast, Fisher et al. (2002) assumed that a single meltwater flood event from Lake Agassiz into the Arctic Ocean resulted in an expansion and thickening of sea-ice as well as in

fresher North Atlantic surface waters.

Despite this cooling, there is no evidence for any significant glacier re-advance in the Woodfjorden area as IRD do not change much (Fig. 7b). Grain-size measurements still illustrate high but steadily decreasing fractions of fine sediments (~7–8.5 µm; Fig. 3a, d), which might reflect continuous meltwater discharge, thus, further deglaciation. Nevertheless, increasing percentages of coarser sediments (~20 µm) are also observed during the

PBO (Fig. 3a, d). These coarser sediments probably result from sea-ice-rafted deposition as sediments >10 µm are interpreted as transported by sea ice (cf., Hebbeln, 2000). During previous time intervals, enhanced percentages of this coarser fraction (~20 µm) were masked because they were diluted by high amounts of meltwater-transported finer sediments (e.g., grain-size distribution at ~13.9 ka BP; Fig. 3d). In contrast, presumed sea-ice-rafted debris is better recognizable during the PBO when the grain-size distribution is less influenced by suspended sediments

derived from meltwater plumes (cf., Fig. 3d). Sea-ice biomarkers exhibit a decreasing trend indicating a moderate sea-ice coverage (Fig. 6b) but peak values of $IP_{25}$ (Fig. 6c) illustrate high amounts of sea-ice diatoms which possibly characterise a sea-ice edge position at the mouth of Woodfjorden. Thus, the deposition of sea-ice-rafted material at the core site seems realistic. Subsequently, maximum accumulation rates of the phytoplankton marker brassicasterol (Fig. 6d) point to algal blooms in spring/summer under open water conditions that persisted

throughout the early Holocene (cf., $PIP_{25}$ values; Fig. 6b and Fig. 7).

At the end of the PBO, *N. labradorica* rapidly recovered (reaching up to ~50 %; Fig. 7c), while *E. clavatum* and *C. reniforme* declined in percentage (Fig. 4f, g). This faunal transition probably reflects the return of the Arctic Coastal Front in the vicinity of the core site, accompanied by rising AW intruding into intermediate water masses in the fjord (cf., sect. 5.2). This hydrographic change was likely a response to the onset of the Holocene Thermal

Maximum (HTM) in Svalbard, at ~10.8 ka BP (cf., Miller et al., 2010), when summer temperatures significantly





exceed modern temperatures due to the higher summer insolation (Fig. 7g; Berger and Loutre, 1991). Peaks in benthic foraminifera flux, with more than 50000 ind. cm$^{-2}$ ka$^{-1}$, as well as enhanced accumulation rates of brassicaserol (Fig. 6d) indicate high food supplies. The observed increase of $\Delta\delta^{13}C$ values (Fig. 7e) implies a high export productivity lasting until ~10.4 ka BP. This may be linked to the presence of AW as a subsurface or even

surface water mass, hence amplifying productivity in the photic zone. The appearance of the thermophilous mussel *Mytilus edulis* at the outer part of Woodfjorden (Reinsdyrflya) supports the assumption of a surface injection of AW at least from ~10.6 ka BP on (9815 ±80 $^{14}$C a BP; Salvigsen, 2002). Several studies around Spitsbergen report about a benthic faunal pattern comparable to the above record during the HTM (Groot et al., 2014; Rasmussen et al., 2012; Skirbekk et al., 2010; Ślubowska-Woldengen et al., 2007). Ślubowska-Woldengen et al. (2008) showed

that *N. labradorica* reached maximum percentages at sites influenced by the WSC or the Svalbard Branch. At the mouth of Woodfjorden, percentages of *I. helenae* (Fig. 4c) rose at the onset of the HTM. This species has been connected to sea-ice algae blooms in various studies (mostly lumped with *I. norcrossi*; e.g., Steinsund, 1994). Because sea-ice coverage shows a decreasing trend (see PIP$_{25}$; Fig. 6b), whereas alkenone-based SSTs reach peak values of ~7 °C (Fig. 6a), it seems more likely that *I. helenae* did not respond specifically to sea-ice algae blooms

but rather to an enhanced productivity (probably due to open waters), as already proposed by Polyak et al. (2002) and Seidenkrantz (2013).

During the HTM, increasing SSTs (based on planktic foraminiferal assemblages) were reconstructed at the western Svalbard continental slope (Hald et al., 2004), south of Spitsbergen at Storfjorden slope (Rasmussen et al., 2014b), and at the western Barents shelf edge (Sarnthein et al., 2003). Lacustrine records from NW-Spitsbergen indicate

maximum temperatures based on alkenones until ~10.5 ka BP (van der Bilt et al., 2016). Carbonara et al. (2016) as well as Rigual-Hernández et al. (2016), report about maximum concentrations of diatoms and coccoliths (mainly warm water species) and inferred a causal strengthened advection of AW during the HTM south of Spitsbergen. Thus, a significant warming in the Svalbard area occurred during this time interval, with temperatures exceeding modern temperatures (cf., alkenone based SST data in Fig. 6a with hydrographic data in Fig. A1 and, e.g., in

Rasmussen and Thomsen, 2014; Rasmussen et al., 2014b).

Iceberg production remained low during the HTM. From ~10.1 ka BP on, IRD percentages were reduced to a minimum (Fig. 7b) – although slightly increasing at around 9 ka BP – indicating a retreat of most glaciers to innermost fjord positions while some large tidewater glaciers (e.g., Monacobreen; Fig. 1c) stayed connected to the fjord system during the early Holocene. Radiocarbon dated beach ridges show a contemporary drop in relative sea

level almost down to the present level (Fig. 7f; Forman et al., 2004), in response to the deglaciation of Spitsbergen with the consequential rebound persisting into the HTM .

Grain-size distribution during the HTM exhibits maxima around 20 µm (Fig. 3a) that may indicate relatively high fluxes of sea-ice rafted debris. In the scenario, some winter sea-ice cover would have been present, although sea-ice biomarker concentration is low during the interval (cf., PIP$_{25}$ minima; Fig. 6b). The PIP$_{25}$ index is based on

sea-ice diatoms and is thought to respond to spring/summer conditions (Müller et al., 2011).

After ~10.1 ka BP another slight shift in the faunal distribution is recognized (Fig. 4): *N. labradorica* declined whereas *C. reniforme* rose again (Fig. 4d, g). This faunal change may be linked to a repeated northward movement of the Arctic Coastal Front away from the core site in concert with submerging AW. Following a significant peak in the benthic foraminifera flux at ~10.1 ka BP, all proxies for productivity unveil a decreasing trend (i.e., $\Delta\delta^{13}C$,





accumulation rates of brassicasterol and benthic foraminifera; Fig. 7e, Fig. 6d) indicating a fading AW influence in the photic zone.

Around 8.8 ka BP, a further coarsening of the sediment is observed (maximum percentages ~45–50 µm, mean: ~15 µm; Fig. 3a, Fig. 7a). This shift in the grain-size distribution may indicate higher current velocities. The
pronounced skewness towards coarse sediments seen in the grain-size distribution point to winnowing, and/or bypass of the fine fraction (cf., grain-size distribution at ~8.8 ka BP in Fig. 3d). The epibenthic species *C. lobatulus* – commonly connected to strong bottom currents (Hald and Korsun, 1997; Hansen and Knudsen, 1995; Steinsund, 1994) – displays slightly risen proportions during this time period (Fig. 4a). Increasing percentages of *B. frigida* s.l. (Fig. 4b) go along with these findings because the species develops preferentially in coarse sediments (Steinsund,
1994). This taxon has also been connected to ice-edge algal blooms (Hald and Steinsund, 1996; Steinsund, 1994). Here however, this seems unlikely; analogue to *I. helenae*, its presence rather corresponds to reduced sea-ice cover conditions during spring/summer, as inferred from declining $PIP_{25}$ values (Fig. 4b, Fig. 6b) (cf., Seidenkrantz, 2013). Apart from the species preference for coarse sediments, the occurrence of *B. frigida* s.l. seems to be mainly controlled by the presence of open waters enabling primary production and high organic matter fluxes to the sea
floor. An enhanced exported production is indeed indicated by increasing $\Delta\delta^{13}C$ values (Fig. 7e) synchronously matching peak percentages of *B. frigida* s.l. at around 8.8 ka BP (Fig. 4b).

### 5.4 Late Holocene (~1.8–0.4 ka BP) – aftermaths of the Neoglaciation

In comparison with the early Holocene, distinct differences in the benthic foraminiferal fauna mark the last ~1400 years of our record. The faunal composition suggests frequent AW incursions, temporarily as bottom water mass:
The *N. labradorica* abundance diminished down to the ~13–22 % range (Fig. 7c), whereas *C. reniforme* exhibits percentages up to ~31 % (Fig. 4g). *C. reniforme* may indicate more saline conditions in bottom waters linked to the presence of AW at the sea floor (Hald and Korsun, 1997; Jennings et al., 2004). Enhanced percentages of the opportunistic species *E. clavatum* (up to ~28 %; Fig. 4f) point to unstable conditions, e.g., high inter-annual variability in environmental conditions. Nonetheless, *I. helenae* reached proportions comparable to those of the
HTM (>5 %; Fig. 4c), reflecting still sufficient food availability. Increasing $\Delta\delta^{13}C$ values (Fig. 7e) characterise this interval, likely responding to the temporary influence of AW in subsurface waters transporting nutrients into the photic zone. Late Holocene seasonally ice-free waters (cf., Fig. 6b) enabled primary productivity despite lower SSTs than those inferred above for the HTM interval (~4 °C; Fig. 6a), very likely due to the decreasing summer insolation and reduced seasonal temperature contrast (cf., Berger and Loutre, 1991; Fig. 7g). Cooling in the Fram
Strait area during the late Holocene is reported at various locations and assigned reduced insolation (e.g., Hald et al., 2004; Müller et al., 2012; Rasmussen et al., 2012; Rigual-Hernández et al., 2016; Ślubowska-Woldengen et al., 2007; Werner et al., 2013). Although sea-ice biomarkers suggests open waters during summer (Fig. 6b, Fig. 7), increased proportions of ~20 µm and ~50 µm grain size fractions (Fig. 3a) may imply the deposition of (winter) sea-ice rafted debris (cf., sect. 5.3).
Parallel to this cooling, a slight increase in IRD percentages indicates enhanced glacier activity with intensified iceberg production (Fig. 7b) in the Woodfjorden area. Furrer et al. (1991) report about advances of Monacobreen into the Liefdefjorden (Fig. 1c) during the Older Subatlantic (~1.2 cal. ka BP; 1315 ±100 $^{14}C$ a BP), whose glacier ice front reaching Lernerøyane, a small archipelago ~6 km off its modern position. Advancing glaciers have also been reported in central Spitsbergen during a similar time period (Baeten et al., 2010; Humlum et al., 2005). The





growth of Monacobreen possibly responded to the neoglacial cooling trend (cf., Porter and Denton, 1967), that led to waxing of Arctic glaciers (e.g., Miller et al., 2010; Solomina et al., 2015, 2016), and is reflected in increased IRD deposition in the present study.

In contrast to conditions inferred above, some other studies state a general warming trend (Bonnet et al., 2010;
D'Andrea et al., 2012; Jernas et al., 2013; Majewski et al., 2009; Spielhagen et al., 2011; Werner et al., 2011) as well as a strengthened influx of AW during the last two millennia in the Svalbard area (Groot et al., 2014; Rasmussen et al., 2012; Sarnthein et al., 2003; Ślubowska et al., 2005). However, as the mid-Holocene is missing in the present record (cf., sect. 4.1), we can only compare the early vs late Holocene. As the late Holocene is marked by cooler conditions than those prevailing during the HTM, a link to the Neoglaciation is assumed, but the
actual trend from the mid- to the late Holocene is unknown.

### 5.5  The role of Atlantic Water in the deglaciation history of the Svalbard-Barents-Sea Ice sheet

Around Svalbard margins, intrusion of AW is generally causally linked to the retreat of glaciers since the late Weichselian (Hormes et al., 2013; Jessen et al., 2010; Klitgaard Kristensen et al., 2013; Skirbekk et al., 2010; Ślubowska-Woldengen et al., 2007; Ślubowska et al., 2005). subsurface inflow of AW into fjords possibly melted
tidewater glaciers from underneath, as proposed by Hormes et al. (2013). Nowadays, submarine melting of tidewater glaciers induced by warm ocean currents is observed in the Arctic, e.g., in Alaska (Bartholomaus et al., 2013; Motyka et al., 2003), Greenland (Holland et al., 2008; Inall et al., 2014; Straneo et al., 2010) and Svalbard (Luckman et al., 2015), as well as in Antarctica, i.e., on the Antarctic Peninsula (Cook et al., 2016; Padman et al., 2012; Wouters et al., 2015).

Numerous studies assume that the intensification of AW inflow along the Svalbard margin occurred relatively synchronously during the Bølling-Allerød interpreting the appearance of *C. neoteretis* (Klitgaard Kristensen et al., 2013; Rasmussen et al., 2007; Ślubowska et al., 2005). In the Laptev Sea, this species has been connected to Atlantic-derived water, from ~15.4 ka BP on (Taldenkova et al., 2010). AW presence is also inferred prior to ~14 ka BP off West Greenland (Sheldon et al., 2016a). Despite the assumed intrusion of relatively warm water masses,
planktic foraminifera assemblages indicate low subsurface temperatures without considerable changes far into the YD, neither along the W-Barents slope (Sarnthein et al., 2003), nor along the W-Svalbard margin (Hald et al., 2004). Thus, a subsurface to bottom inflow of relative warm AW is assumed during the Bølling-Allerød, while surface waters remained cold.

The YD cold event has been studied extensively. This cold reversal was most likely initiated by a massive glacial
lake drainage into the Arctic Ocean that reduced the Atlantic Meridional Overturning Circulation (Broecker, 2006; Broecker et al., 1989; Fahl and Stein, 2012; Murton et al., 2010; Stein et al., 2012; Tarasov and Peltier, 2005). Nonetheless, the influence of AW is seen at numerous locations around Svalbard, at least during the early YD and possibly temporarily (e.g., Rasmussen et al., 2007; Ślubowska et al., 2005; this study, see Fig. 7c, d). A shift of the Arctic Coastal Front towards the study areas at the northern Svalbard margin characterises the situation during
the YD. The proximal position of this oceanic front is linked to an initial AW intrusion into intermediate waters of Woodfjorden providing higher food supplies (Fig. 7e) while contributing to the retreat of tidewater glaciers (Fig. 7a, b).

During the early Holocene, the Nordic Seas, the Fram Strait and Barents Sea were characterized by a high AW inflow (e.g., Bauch et al., 2001; Carbonara et al., 2016; Groot et al., 2014; Müller and Stein, 2014; Rigual-



Hernández et al., 2016; Risebrobakken et al., 2011; Telesiński et al., 2015; Werner et al., 2013, 2016), coinciding with a general warming trend in the entire Svalbard region – especially after the PBO (e.g., Hald et al., 2004; Rasmussen et al., 2012; Sarnthein et al., 2003; Skirbekk et al., 2010; Ślubowska et al., 2005; this study, see Fig. 6a). This trend is in agreement with the early Holocene warming observed in the entire North Atlantic (Marchal et al., 2002).

The interruption of the warming trend during the PBO occurred almost synchronously around Svalbard but is more pronounced in northern locations (Rasmussen et al., 2014b; Skirbekk et al., 2010; Ślubowska et al., 2005; this study, see Fig. 4, Fig. 6) than along the (south-)western margin of Svalbard (Rasmussen et al., 2012, 2014b; Ślubowska-Woldengen et al., 2007). Accordingly, it seems likely that the PBO was triggered by a vast meltwater flood from the Mackenzie River area into the Arctic Ocean (Fisher et al., 2002), than by a meltwater discharge from the Fennoscandian Ice Sheet into the Nordic Seas as assumed earlier (cf., Björck et al., 1996). Thus, northern sites may have been more strongly affected by the resulting colder and fresher surface waters and the expansion of a thicker pack ice, which concured to force AW to sink deeper in the water column.

Nevertheless, as discussed above, AW was present at the northern margin of Spitsbergen during the last two millennia, although its influx was weaker and its position possibly deeper in the water column, when compared to HTM conditions (Fig. 7c). Various studies report similar oceanographic conditions during this time interval (e.g., Hald et al., 2004; Rasmussen et al., 2014b; Skirbekk et al., 2010; Ślubowska-Woldengen et al., 2007; Ślubowska et al., 2005). Contrary to the HTM and the YD, there are indications of glacier advances regardless of the AW advection during the late Holocene (Fig. 7a, b), possibly resulting from the lower summer insolation of the interval (cf., Berger and Loutre, 1991; see Fig. 7g). Additionally, the AW incursion into Woodfjorden as a relatively deep water mass may have been of minor influence on glacier fronts as tidewater glaciers may only have been present in the shallower inner parts of the fjord system during the late Holocene (cf., Furrer et al., 1991; Salvigsen and Høgvard, 2006). Hence, the glacier dynamics were possibly decoupled from AW advection during the late Holocene.

## 6 Conclusions

Multi-proxy analyses of sediment core GeoB10817-4 from the mouth of Woodfjorden (northern Spitsbergen) enabled us to document ocean–ice interactions since the deglaciation (since ~15500 years) at the northern margin of Svalbard, providing insights into glacier activity as well as into bottom, intermediate and surface water conditions, including sea ice coverage.

- The deglaciation period (until ~12.7 ka BP; excluding the Younger Dryas) was marked by a disintegrating Svalbard–Barents–Sea Ice Sheet, as illustrated by high calving rates and enhanced meltwater inputs. Cold surface waters were covered by extensive sea ice, while early intrusions of Atlantic Water as a bottom water mass occurred (Fig. 8a).
- During the early Younger Dryas (~12.7–12.4 ka BP), Atlantic Water was still advected as an intermediate water mass, probably connected to a coastward shift of the Arctic Coastal Front. Its intrusion was accompanied by warmer surface waters and high iceberg melting rates, which may have been linked to the retreat of some tidewater glaciers towards the fjord heads (Fig. 8b). During the late Younger Dryas (~12.4–11.7 ka BP), surface waters cooled while sea ice extended and Atlantic Water temporarily penetrated into bottom waters.



- Conditions improved during the early Holocene (~11.7–7.8 ka BP), with an increasing subsurface Atlantic Water inflow, and were briefly interrupted during the Preboreal Oscillation. This deterioration occurred almost synchronously around Svalbard, but was more pronounced at the northern margin of Svalbard than in the south. During the succeeding Holocene Thermal Maximum, sea surface temperatures reached their maximum surpassing modern temperatures. Productivity flourished and sea ice coverage declined. Most glaciers retreated to innermost fjord positions during this interval. Only single tidewater glaciers survived (especially Monacobreen) (Fig. 8c).

- The early vs late Holocene depict distinct environmental conditions. Productivity gradually increased in the course of the late Holocene (~1.8–0.4 ka BP) while Atlantic Water temporarily sank from intermediate to bottom waters. Even though sea surface temperatures were lower than during the early Holocene, sea ice coverage was reduced. High iceberg production possibly responded to some late Holocene glacier advance (Neoglaciation) in response to low summer insolation (Fig. 8d).

- From the deglaciation on, the presence of Atlantic Water – in bottom as well as in intermediate waters – is connected to retreating glaciers at the northern Svalbard margin, while during the late Holocene, glaciers might have grown despite relatively constant inflow of Atlantic Water (possibly below surface).

- The Holocene Thermal Maximum includes – apart from a few differences (e.g., considerable higher summer insolation, dropping relative sea level due to the glacio-isostatic rebound) – numerous similarities to a future warming: Sea surface temperatures were most likely warmer than today, resulting in a reduced sea ice coverage and consequential open waters that enabled high primary production. Already nowadays, reduced sea ice promotes increasing phytoplankton blooms (Arrigo and van Dijken, 2011, 2015). During the Holocene Thermal Maximum, subsurface intrusion of warm Atlantic Water contributed to melting and retreat of glaciers to inner fjord positions; a phenomenon that is also observed today, e.g., at Svalbard´s tidewater glaciers (Luckman et al., 2015). Thus, environmental conditions during the Holocene Thermal Maximum are well comparable with the already ongoing global warming; at least for the Svalbard region.

**Data availability**

Supplementary data are available on PANGAEA: https://www.pangaea.de/ (will be uploaded after revision).

**Author contributions**

M. Bartels and D. Hebbeln designed the study. M. Bartels performed the faunal analyses, created the figures and interpreted all data. J. Titschack created the base maps in Fig. 1 (b and c), performed the particle size measurements and CT analyses as well as respective data processing. K. Fahl carried out the biomarker analyses. M.-S. Seidenkrantz revised the identification of the foraminiferal species and contributed to their interpretation. D. Hebbeln, C. Hillaire-Marcel and R. Stein contributed significantly to the discussion of the data. M. Bartels prepared the manuscript with contributions from all co-authors.

The authors declare that they have no conflict of interest.





**Acknowledgements**

This project was supported by the *Deutsche Forschungsgemeinschaft* (DFG) through the International Research Training Group "Processes and impacts of climate change in the North Atlantic Ocean and the Canadian Arctic" (IRTG 1904 *ArcTrain*). We would like to thank the captain and crew of RV *Maria S. Merian* as well as the scientists on board during the cruise MSM02/03 in 2006 for retrieving the samples. Following persons are acknowledged for the performance of measurements: Janet Rethemeyer, University of Cologne, and Lukas Wacker, ETH (Zürich), for radiocarbon measurements as well as Birgit Meyer and Henning Kuhnert, MARUM (Bremen), for stable isotope measurements, and Walter Luttmer, AWI (Bremerhaven), for technical assistance. Further we thank Hyunyung Boo, McGill University (Montreal), as well as Nele Lamping, Jan Unverfärth and Marco Wolsza, MARUM (Bremen), for sample processing. Sample material was provided by the GeoB repository at MARUM, Bremen. Klinikum Bremen-Mitte is gratefully acknowledged for providing their facilities for the performed computed tomographies. Arne-Jörn Lemke and Christian Timann, Gesundheit Nord (Bremen), are thanked for performing the CT scans and their support during the measurements.





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



**Table 1: AMS radiocarbon measurements and calibrated ages applying the Marine13 calibration curve (Reimer, 2013) and an averaged regional ΔR = 98±37 years (calib.qub.ac.uk). All measurements carried out at ETH Zürich apart from the measurements at 2.5 and 270.5 cm (CologneAMS, University of Cologne). *Radiocarbon age from 257 cm was excluded from the age model (as discussed in sect. 4.2).**

| Lab ID | depth | dated material | $^{14}$C age | | cal. age (2σ) [a BP] | | |
|--------|-------|----------------|--------------|-----|-----|-----|--------|
| | [cm] | | [a] | ± | min | max | median |
| COL2571.1.1 | 2.5 | mixed benthic foraminifera | 879 | 31 | 307 | 496 | **420** |
| ETH-58439 | 27.5 | mixed benthic foraminifera | 2155 | 45 | 1499 | 1795 | **1630** |
| ETH-61277 | 57.5 | *N. labradorica* | 7845 | 65 | 8035 | 8360 | **8216** |
| ETH-64606 | 67 | *N. labradorica* | 8815 | 60 | 9217 | 9519 | **9384** |
| ETH-61278 | 77 | *N. labradorica* | 9395 | 75 | 9846 | 10339 | **10114** |
| ETH-61279 | 122 | *N. labradorica* | 9980 | 75 | 10613 | 11082 | **10837** |
| ETH-61280 | 157 | *N. labradorica* | 10230 | 75 | 10873 | 11306 | **11127** |
| ETH-58440 | 182.5 | mixed benthic foraminifera | 10710 | 70 | 11445 | 12276 | **11918** |
| ETH-64607 | 212 | *N. labradorica* | 11120 | 80 | 12283 | 12726 | **12559** |
| ETH-58441 | 237 | mixed benthic foraminifera | 11810 | 70 | 13000 | 13363 | **13197** |
| ETH-64608 | 257 | mixed benthic foraminifera | 11610* | 120 | | age reversal | |
| COL2572.1.1 | 270.5 | mixed benthic foraminifera | 13391 | 53 | 15167 | 15648 | **15391** |









**Fig. 1: a) Average annual (1955–2012) sea surface temperatures (SST) of the North Atlantic (World Ocean Atlas/Ocean Data View; Locarnini et al., 2013; Schlitzer, 2015) White rectangle: Svalbard area as shown in (b). b) Main ocean currents in the Svalbard area. Red arrows: Atlantic Water (WSC = West Spitsbergen Current; SB = Svalbard Branch). Blue arrows: Arctic Water (ESC = East Spitsbergen Current; CC = Coastal Current). Orange dot marks study location (GeoB108*17-4*). Yellow dots mark other core sites mentioned in the text: 1. NP94-51 (Jernas et al., 2013; Koç et al., 2002; Ślubowska-Woldengen et al., 2007; Ślubowska et al., 2005); 2. MSM5/5-712-1** (Müller et al., 2012; Spielhagen et al., 2011; Werner et al., 2011, 2013, 2014)**; 3. NP05-11-21 (Jernas et al., 2013; Rasmussen et al., 2014b; Skirbekk et al., 2010); 4. JM98-845 (Rasmussen et al., 2012); 5. MD99-2304 (Hald et al., 2004); 6. MD99-2305 (Hald et al., 2004); 7. JM02-440 (Ślubowska-Woldengen et al., 2007); 8. HR3 (Majewski et al., 2009); 9. JM02-460 (Rasmussen et al., 2007); 10. 23258 (Sarnthein et al., 2003). White rectangle: Study area as shown in (c). c) Woodfjorden area with eponymous fjord and adjacent fjords. EB = Emmabreen; IB = Idabreen; SB = Seligerbreen; LØ = Lernerøyane. Orange dot marks study location (GeoB108*17-4*). The bathymetric metadata and Digital Terrain Model data products in (b) and (c) have been derived from the EMODnet Bathymetry portal (http://emodnet-bathymetry.eu/). Topographic data in (c): © Norwegian Polar Institute (Norwegian Polar Institute, 2014).**







Fig. 2: a) CT scan of the core GeoB10817-4. Left to right: orthogonal profile; interpreted image (clasts and bioturbation, s. legend); clast-size distribution (0–20 vol. % of clasts: blue to red, respectively). Yellow rectangle marks the disturbed section also shown in (b). b) CT images (orthogonal and interpreted) of the disturbed section (yellow rectangle; definition of boundaries: see sect. 4.1 and Fig. A3 in the appendix). c) Age–depth plot for sediment core GeoB10817-4: Diamonds show calibrated (median) ages with error bars (Table 1); solid black line shows linear interpolation excluding an outlier at 257 cm. Blue solid line shows corresponding sedimentation rates.





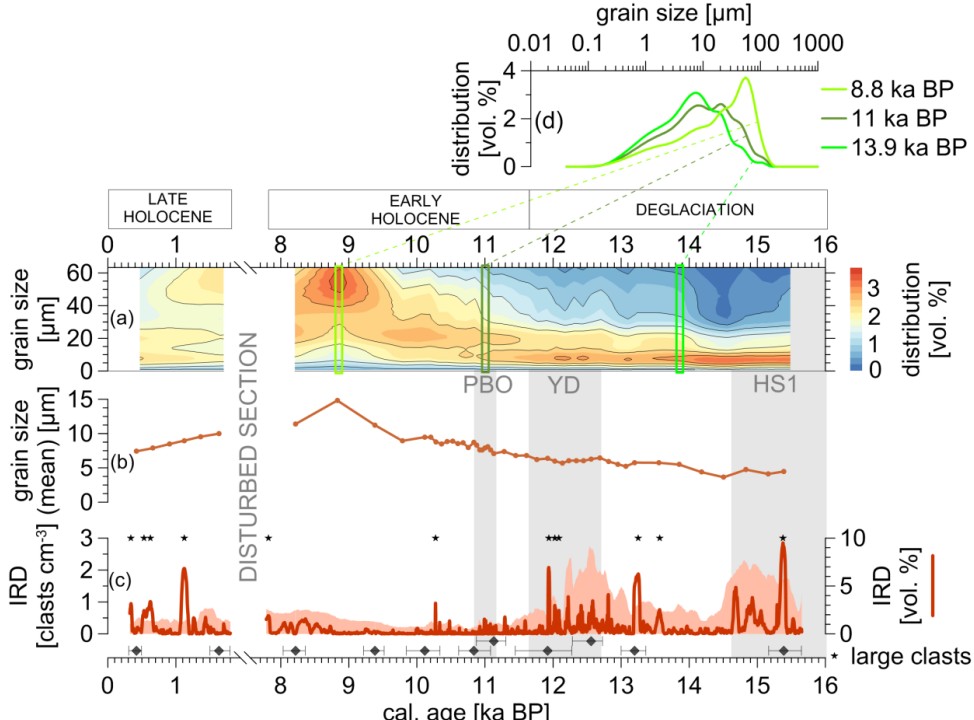

**Fig. 3: a) Grain-size distribution [vol. %] of fine siliciclastic sediments (0–63 µm) derived from laser diffraction particle size analyses. Green rectangles: selected grain-size distributions as shown in (d). b) mean grain size [µm] of siliciclastic sediments. c) Ice rafted debris (IRD): volume percentage (solid red line) and clasts per cm³ (reddish shading) derived from CT analysis. Stars mark peaks in vol. % corresponding to single large clasts. Grey vertical shadings indicate cold periods: Preboreal Oscillation (PBO), Younger Dryas (YD) and Heinrich Stadial 1 (HS1). Dark grey diamonds: calibrated radiocarbon dated depths with error ranges. d) selected grain-size distributions [vol. %] at ~8.8 ka (light green line), ~11 ka (dark green line) and ~13.9 ka BP (green line).**



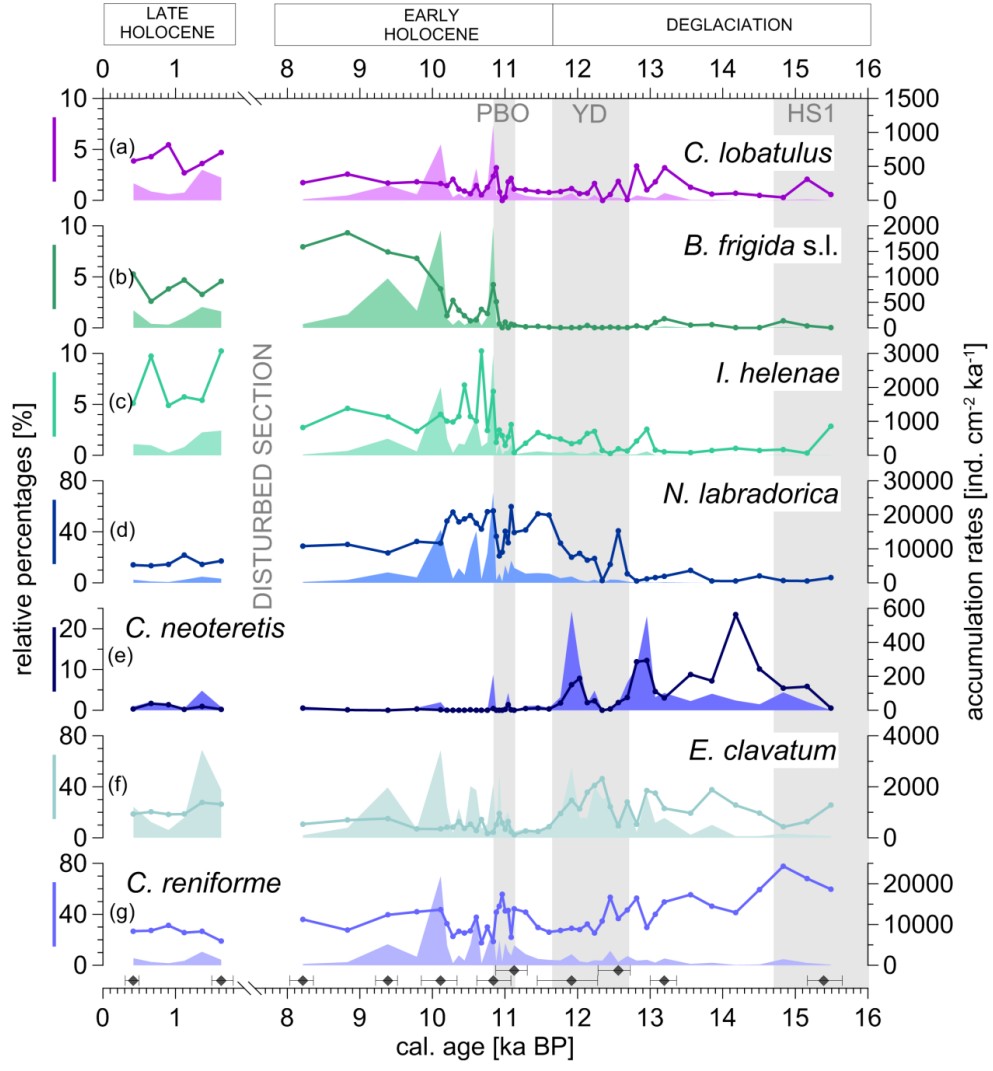

Fig. 4: Relative percentages (lines) and accumulation rates (shadings) of dominant calcareous benthic foraminifera.
a) *Cibicides lobatulus*. b) *Buccella frigida* s.l. c) *Islandiella helenae*. d) *Nonionellina labradorica*. e) *Cassidulina neoteretis*.
f) *Elphidium clavatum*. g) *Cassidulina reniforme*. Grey vertical shadings: s. Fig. 3. Dark grey diamonds: calibrated
radiocarbon ages with error ranges.




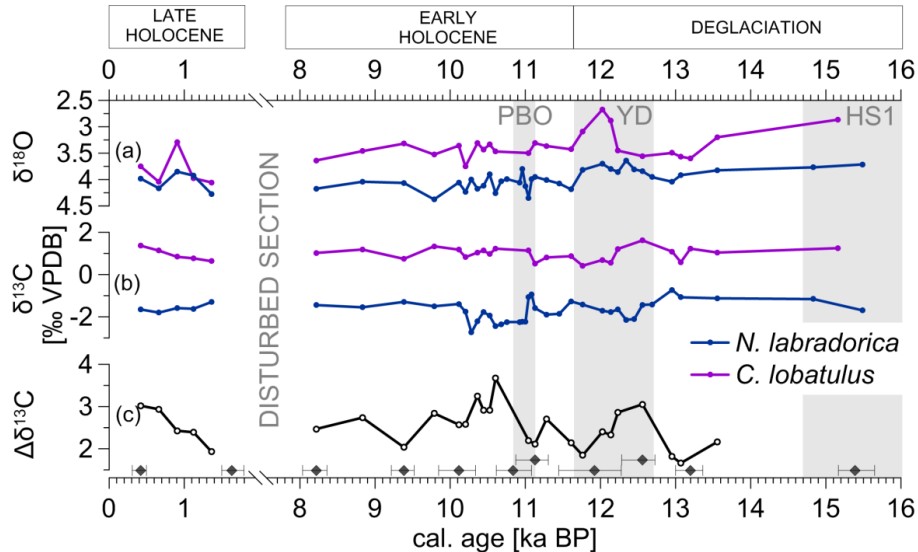

**Fig. 5:** a) Stable oxygen isotopes (corrected for vital effects and the global ice effect; s. sect. 3.5) and b) stable carbon isotopes of the epibenthic species *Cibicides lobatulus* (purple line) vs the endobenthic species *Nonionellina labradorica* (blue line). c) Difference between epi- and endobenthic stable carbon isotope compositions. Grey vertical shadings: s. Fig. 3. Dark grey diamonds: calibrated radiocarbon ages with error ranges.





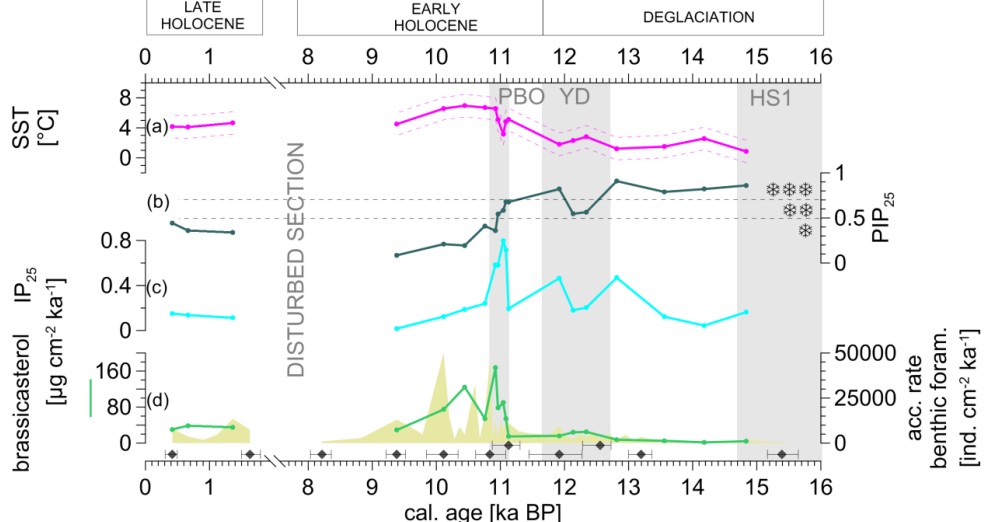

**Fig. 6: a) Sea surface temperature (SST) with standard error (±1.5 °C) derived from alkenone unsaturation index ($U_{37}^K$) (Brassell et al., 1986; Müller et al., 1998). b) Phytoplankton (brassicasterol)-IP$_{25}$ index (PIP$_{25}$). 0.3 to 0.5: reduced sea ice cover (❋), 0.5 to 0.7: seasonal sea ice cover (❋❋), >>0.7: extended to perennial sea ice cover (❋❋❋) (Müller et al., 2011). c) Accumulation rate of sea ice biomarker IP$_{25}$. d) Accumulation rates of phytoplankton biomarker brassicasterol (green line) and of benthic foraminifera (greenish shading). Grey vertical shadings: s. Fig. 3. Dark grey diamonds: calibrated radiocarbon ages with error ranges.**



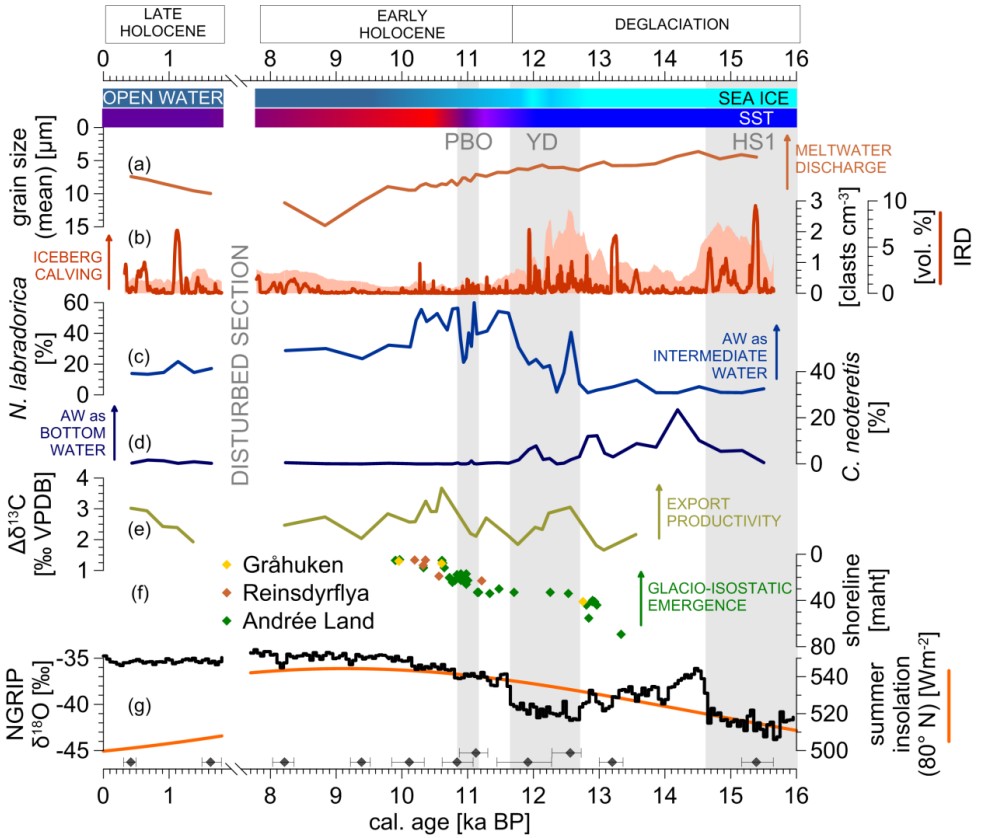

**Fig. 7:** Blueish horizontal bar on top symbolises sea-ice coverage based on biomarker data (PIP$_{25}$ index). Red–blue horizontal bar symbolises (qualitative) sea surface temperatures (red = warm; blue = cold) based on biomarker data (alkenones). a) Mean grain size of siliciclastic (terrigenous) sediments: low values illustrate deposition of suspended sediment from meltwater plumes (note descending y-axis). b) Volume percentage of ice-rafted debris (solid red line) and number of clasts per cm³ (reddish shading) signifying glacier activity (iceberg calving). c) Relative percentages of *N. labradorica*. High percentages: Atlantic Water intrusion into intermediate waters delivering nutrients into the photic zone. Note the almost parallel increase in export production as illustrated in (e). d) Relative percentages of *C. neoteretis*. High percentages: Atlantic Water advection into bottom waters. e) Offset of epi- and endobenthic δ¹³C values indicating export productivity. f) Radiocarbon dated beach ridges in the Woodfjorden area (Forman et al., 2004) illustrating glacio-isostatic emergence (in metres above high tide [maht]; note descending y-axis). Colours mark different locations as indicated in the legend (Fig. 1c). g) Stable oxygen isotopes from NGRIP ice core (black line; Rasmussen et al., 2014a) and June/July insolation at 80° N (orange line; Berger and Loutre, 1991). Grey vertical shadings: s. Fig. 3. Dark grey diamonds: calibrated radiocarbon ages with error ranges







Fig. 8: Environmental conditions in the Woodfjorden during (a) the deglaciation, (b) the Younger Dryas, (c) the early Holocene and (d) the late Holocene. Black bars mark location of sediment core GeoB108*17-4*. Glacier retreats or advances are marked by turquoise arrows. Yellow circles symbolise insolation changes (cf., Berger and Loutre, 1991).

5    Crossed circle in (b) signifies easterlies (blowing perpendicular to the profile) triggering a northward Ekman transport of the surface waters which entailed an upwelling of intermediate waters (i.e., AW) (cf., Lind and Ingvaldsen, 2012).



**Appendices**

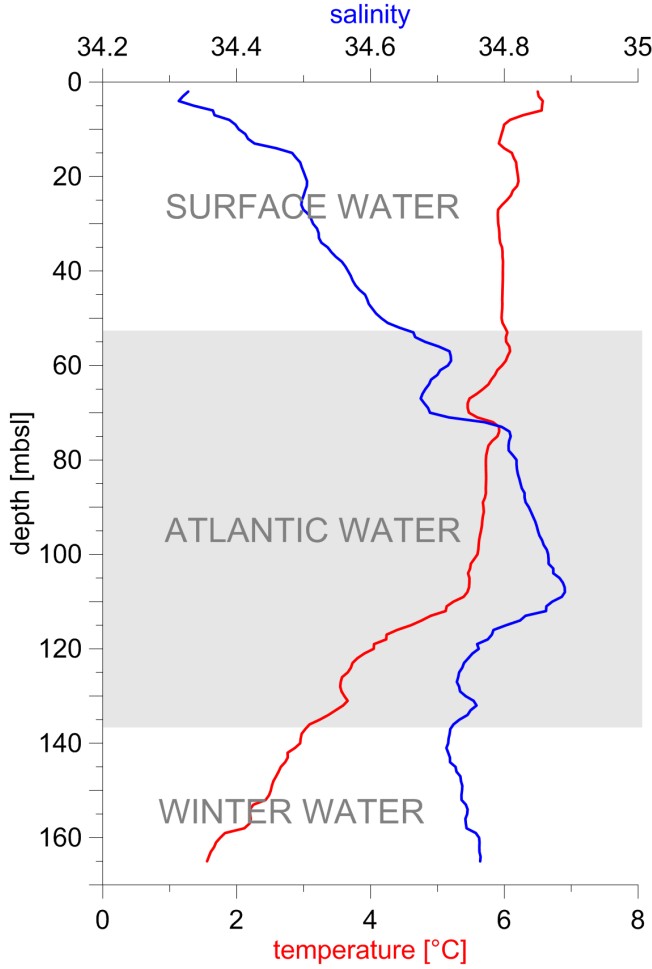

**Fig. A1: Temperature and salinity profile at station MSM02/03-666-3 (same location as the studied sediment core
GeoB10817-4; see sect. 3) derived during cruise MSM02/03 applying a Sea Bird CTD sensor as described in the cruise
report (Lherminier et al., 2009). The profile illustrates subsurface Atlantic Water inflow at the core site (grey shading),
framed by less saline Surface Water on top and colder Winter Water at the bottom. Winter Water forms locally by
convection processes during autumn/winter (cf., Cottier et al., 2010). Temperature (>3 °C) and salinity boundaries
(>34.65) following Cottier et al. (2005).**





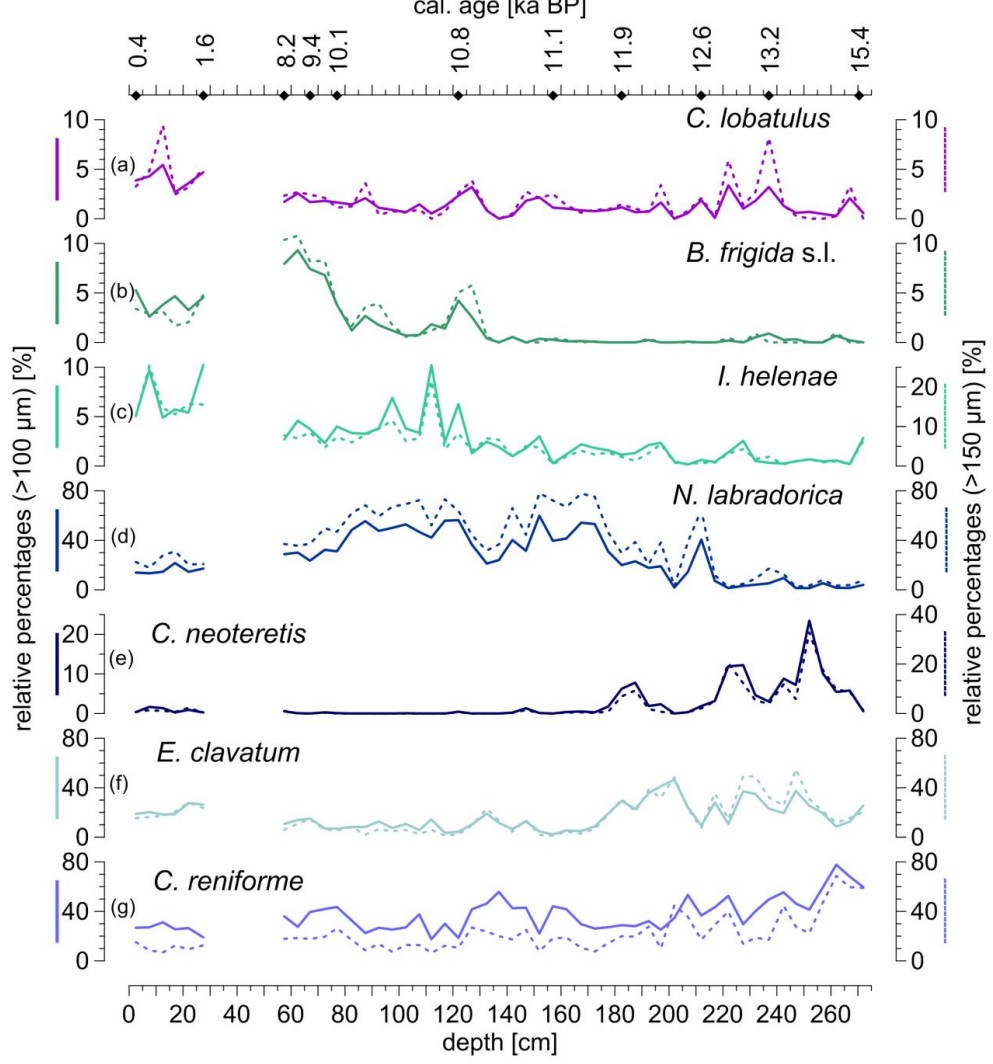

**Fig. A2: Relative percentages of the dominant calcareous benthic foraminifera. Comparison of fraction >100 μm (solid lines) vs fraction >150 μm (dashed lines). a)** *Cibicides lobatulus.* **b)** *Buccella frigida* **s.l. c)** *Islandiella helenae.* **d)** *Nonionellina labradorica.* **e)** *Cassidulina neoteretis.* **f)** *Elphidium clavatum.* **g)** *Cassidulina reniforme.* **Note similar pattern of the faunal distributions. The highest offset can be recognised for** *N. labradorica* **and** *C. reniforme* **because the majority of** *N. labradorica* **appears in the >150 μm fraction whereas the majority of** *C.reniforme* **appears in the >100 μm fraction. Black diamonds: calibrated radiocarbon ages.**





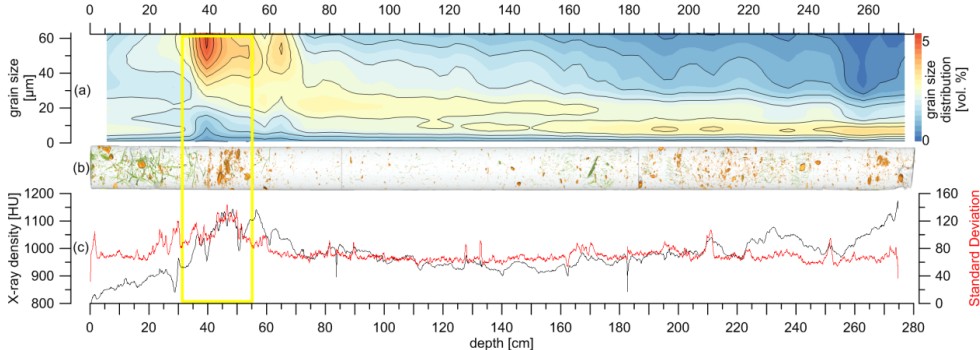

**Fig. A3: a) Grain size distribution of fine siliciclastic sediments (0–63 μm). b) Interpreted CT analysis showing bioturbation (green) and clasts (orange). c) X-ray density (black line) in Hounsfield units (HU) with standard deviation (red line) derived from CT analysis. Yellow rectangle marks disturbed section. Note coarsening upwards of fine sediments (a), high accumulation of clasts (b) and increased x-ray density as well as respective standard deviation (c) in the disturbed section.**





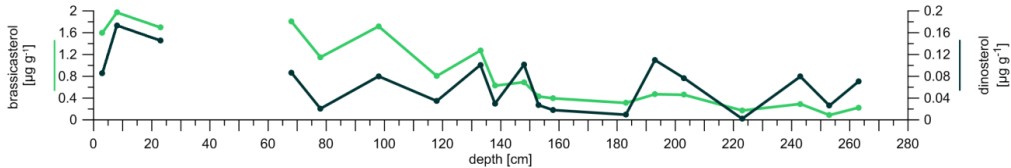

**Fig. A4: Phytoplankton biomarker [µg per g Sediment]: Brassicasterol (light green line) and dinosterol (dark green line). Note almost parallel trend with brassicasterol showing one magnitude higher values than dinosterol.**





**Table A1: Lists of benthic foraminiferal species appearing in sediment core GeoB10817-4**

**Agglutinating species**

*Adercotryma glomerata* (Brady, 1878)

*Ammodiscus* sp.

*Bathysiphon rufum* de Folin, 1886

*Bathysiphon* spp.

*Cribrostomoides* spp.

*Cuneata arctica* (Brady, 1881)

*Eggerella* spp.

*Hippocrepinella* sp.

*Lagenammina* spp.

*Miliammina spp.*

*Portatrochammina bipolaris* Bronnimann & Whittaker, 1980

*Portatrochammina karica* (Shchedrina, 1946)

*Portatrochammina* spp.

*Recurvoides turbinatus* (Brady, 1881)

*Reophax* spp.

*Spiroplectammina biformis* (Parker & Jones, 1865)

*Textularia earlandi* Phleger, 1952

*Textularia* spp.

*Textularia torgata* Parker, 1952

Textulariina

*Trochammina astrifica* (Rhumbler, 1938)

*Trochammina* spp.

**Calcareous species**

*Astrononion gallowayi* Loeblich & Tappan, 1953

*Bolivina pseudopunctata* Höglund, 1947

*Buccella frigida* s.l. (*B. frigida* s.s. (Cushman, 1922) and *B. tenerrima* (Bandy, 1950))

*Buccella* spp.

*Bulimina* c.f. *alazanensis/rostrata*

*Cassidulina neoteretis* Seidenkrantz, 1995

*Cassidulina reniforme* Nørvang, 1945

Cassidulininae

*Cibicides lobatulus* (Walker & Jacob, 1798)

Cibicidinae

*Cornuspira involvens* (Reuss, 1850)

*Cornuspira* spp.

*Dentalina* spp.

*Discorbinella araucana* (d'Orbigny, 1839)

*Elphidium albiumbilicatum* (Weiss, 1954)





*Elphidium asklundi* Brotzen, 1943

*Elphidium bartletti* Cushman, 1933

*Elphidium clavatum* Cushman, 1930 (formerly named *Elphidium excavatum* (Terquem, 1875) forma *clavata* Cushman, 1930; see Darling et al. (2016))

*Elphidium hallandense* Brotzen, 1943

*Elphidium spp.*

*Elphidium williamsoni* Haynes, 1973

*Epistominella* spp.

*Fissurina orbignyana* Seguenza, 1862

*Fissurina* spp.

*Glandulina ovula* d'Orbigny, 1846

*Globobulimina auriculata* (Bailey, 1894)

*Globobulimina* spp.

*Globobulimina turgida* (Bailey, 1851)

*Globulina* spp.

*Haynesina* spp.

*Islandiella helenae* Feyling−Hanssen & Buzas, 1976

*Islandiella islandica* (Nørvang, 1945)

*Islandiella norcrossi* (Cushman, 1933)

*Islandiella* spp.

*Lagena distoma* Parker & Jones, 1864

*Lagena laevis* (Montagu, 1803)

*Lagena striata* (d'Orbigny)

*Melonis barleeanus* (Williamson, 1858)

*Melonis* spp.

Miliolida

*Miliolinella* spp.

*Nodosaria spp.*

*Nonionella auriculata* Heron-Allan & Earland, 1930

*Nonionella turgida* (Williamson, 1858)

*Nonionellina labradorica* (Dawson, 1860)

*Oolina hexagona* (Williamson, 1848)

*Pullenia quinqueloba* (Reuss, 1851)

*Quinqueloculina seminula* (Linné, 1758)

*Quinqueloculina spp.*

*Quinqueloculina stalkeri* Loeblich & Tappan, 1953

*Robertina* spp.

*Rosalina* spp.

Rotaliina

*Stainforthia loeblichi* s.l. (*S. loeblichi* s.s. (Feyling−Hanssen, 1954) and *S. concava* Höglund, 1947)

*Stainforthia spp.*



*Trifarina angulosa* (Willamson, 1858)

*Trifarina fluens* (Todd, 1947)

*Triloculina* spp.

*Triloculina trihedral* Loeblich & Tappan, 1953

*Uvigerina mediterranea* Hofker, 1932

*Uvigerina* spp.