# Peer review of "Atlantic Water advection vs glacier dynamics in northern Spitsbergen since early deglaciation"

_Climate of the Past, 2017_

## Short Comment (SC1) · 21 Apr 2017

Dear authors,

Your manuscript presents interesting new multiproxy record of paleoceanographic changes from the northern Svalbard shelf area. This is an important contribution adding a valuable new palaeoclimate data point in a remote area. The observed changes since deglaciation matches well with our study in Storfjordrenna. For instance, in your manuscript (page 12, lines 21-31) you suggest that the observed coarsening of the sediment during Allerød interstadial might be connected to the deposition of finer fraction directly in the vicinity of the glacier due to particles flocculation. According to Lacka et al. (2015) the sediment deposition during deglaciation of Storfjordrenna occurred due to suspension settling from sediment-laden plumes and ice-rafting debris

visible as bimodal composition of grain-size distribution. It seems that for the same period you also observed similar sediment composition, with domination of silt-clay fraction and large contribution of IRD (Fig. 3). During Younger Dryas you report the temporarily reduced sea-ice cover and maximum amounts of IRD in Woodfjorden (page 13, lines 19-36). Contrary conditions were observed in southern Svalbard (Lacka et al., 2015) for the onset of Younger Dryas, with perennial sea-ice coverage leading to decreased ice-rafting. It would be interesting to compare those data. On page 14, lines 2-5 you have also noticed the shift from relatively stable to unstable conditions during YD and you refer to study after Bakke et al. (2009) and Pearce et al. (2013), however we have observed similar unstable oceanographic conditions also in the southern Svalbard area (Lacka et al., 2015). We have concluded that YD was not uniformly cold and that at least a number of warmer spells occurred on southern Svalbard. During Holocene Thermal Maximum you have noticed that the iceberg production remained low (page 16, lines 26-29), however at around 9 ka BP and 8.8 ka BP IRD percentages increased. The same elevated values were observed in southern Svalbard (Lacka et al., 2015) during this period. We have explained it as minor cooling, leading to seasonal sea ice formation and beach sediment transport by shore ice. The last suggestion concerns the weaker influx of AW during late Holocene and its deeper position in the water column of Woodfjorden (page 19, lines 14-16). According to our record (Lacka et al., 2015) after 3.7 ka BP AW in Storfjordrenna was only sporadically present at the surface.

I hope you will find those comments useful. The paper I refer to is: Lacka, M., Zajaczkowski, M., Forwick, M., Szczucinski, W.: Late Weichselian and Holocene paleoceanography of Storfjordrenna, southern Svalbard, Climate of the Past, 11, 587–603, 2015

Kind Regards, Magdalena Lacka, PhD mlacka@iopan.gda.pl, Institute of Oceanology, Polish Academy of Sciences, 81-712 Sopot, Poland

---

## Referee Comment (RC1) · Anonymous Referee #1 · 10 May 2017

Review of manuscript cp-2017-53 by Bartels et al. "Atlantic Water advection vs glacier dynamics in northern Spitsbergen since early deglaciation" submitted to Climate of the Past.

The manuscript deals with a marine core record from the mouth of Woodfjorden, northern Svalbard margin from 171 m water depth and 275 cm long. The core comprise the time intervals c. 15.5-7.8 ka and c. 1.8-0.4 ka. The proxies are benthic foraminiferal fauna analysis, stable isotopes, grain size, IRD analysis, alkenone surface temperature and sea ice proxy IP25 and brassicasterol (PIP25 index). The purpose is to study the inflow of Atlantic Water and its link to the behaviour of local glaciers during different climatic phases.

The manuscript is overall well-written and well-structured and of potential interest.

However, it has also some issues, which must be considered before publication. The major issues are that this is a very local study that mainly compares to other local records from the Svalbard margin. Also, changes in reservoir age through time is not dealt with or discussed and in addition are resulting in erroneous correlation to the ice core records. Furthermore, the results rely to a great deal on the ecology of benthic foraminiferal species, which is not discussed in any detail with just a few references and without revealing any details about species from these references. Below I outline these and other concerns in more detail:

The present ecology of foraminifera is very sporadically discussed or described and with only few references. Actually, studies such as by Hald and Korsun, 1997 and Korsun and Hald, 2000 of the distribution of living benthic foraminifera in Svalbard fjords have characterized species in terms of 'ice-proximal', 'ice-distal' (which are useful terms for the present study from a fjord affected by glaciers) and also related them to influx of AW and meltwater to the fjords. There are references to foraminiferal studies from Iceland – I would think living foraminifera in Svalbard fjords today and from East Greenland and the Arctic Ocean would be more relevant (e.g. Ahrens et al., 1997; Newton and Rowe, 1995; Jennings refs from the Greenland margin; Wollenburg refs from north of Svalbard). Often the ecology of the foraminifera is presented as cf. (ref) (e.g. p.14, line 18, p. 15, line 10, p. 17, line 12). I would like to know what these references contain – not having to read the whole publication to find out what is meant here. There is in general too many 'cf.'s' in this manus – one or two is acceptable, but not more. It has to be written out clearly what exactly in all these published papers that is referred to here.

'Abstract': line 6: "spanning the last c. 15,000 years". Your record has a six thousandyear gap - this should be made clear from the start. Also, please mention the core name, length and water depth in the abstract.

'Introduction': p. 4, lines 8-11. This is a bit of over statement: the records you refer to are also multiproxy comprising planktic and benthic stable isotopes and planktic and

benthic foraminiferal faunas, diatoms, various grain sizes and IRD and are most often also of much higher resolution than your record. Better to state that your record is in a more ice proximal setting than the others, and that you can add low-resolution biomarkers (with one or two data points per 1000 years) and a different method for IRD-analysis in fairly high resolution.

p. 5, line 11: give the time interval range between the 5 cm samples and the biomarker samples (also distance between the latter) and the range of time within the 1-cm samples.

Chapter 3.1 'Computer tomography': must be written clearer: what is the resolution of counts of the IRD > c. 1 mm and the volume %? I am also unfamiliar with this method and suggest that a simple count of IRD is performed and shown. This would not be a great effort as there are sieved residues from foraminiferal samples and I suggest IRD > 1mm and also 0.5-1 mm be counted for comparison with the computer analyses and for the sake of comparison with published IRD counts.

p. 6. 'Grain size measurements'. Why are sortable silt data not presented and discussed? The manuscript mainly discusses the silt size fraction in terms of ice rafting from sea ice and/or supplied from glaciers. This size fraction can come from glaciers and sea ice yes – but can be current sorted as well. To me the coarsening from the Holocene onwards is because of current sorting – bottom currents are only mentioned on p. 17, line 4. I cannot see why presence of coarser silt should indicate sea-ice rafting, when definitely there is bottom current activity as indicated by certain benthic foraminiferal species. This may also explain the low sea-ice biomarker concentration mentioned on p. 16, lines 32-35. The maxima in 20 my could be more due to current sorting, than ice rafting in my opinion.

p. 6, lines 36-37: "several specimens show signs of dissolution or transport" – any particular interval or throughout the record? "Small fragments were not counted since these specimens might be allochtonous" – still, it would be good to see the % fragmen-

СЗ

tation data and how they compare to species distribution patterns

p. 7, line 20: "the endobenthic foraminifera Nonionellina labradorica" – there is no documentation at all in this manuscript about the life mode of N. labradorica – is it shallow infaunal or deep infaunal? What kind of food does it require? Fresh or partly degraded, refractory? Or? Any vital effects on 13C? Later the differences in 13C values between C. lobatulus (epibenthic, ref.?) and N. labradorica are used as an indication of export productivity. You have a low-resolution brassicasterol flux record (that do not match the benthic foraminiferal AR) – how does your 'export production delta13C' record compare to the brassicasterol flux and benthic AR? (they are not shown in the same figure). N. labradorica is a deep infaunal species, and changes in 13C may well be that the species moves up or down in the sediment in search for food, more than a direct signal of the amount of food.

p. 9, line 18. I believe is it not necessary to write 'BP' or 'cal.' I suggest to make a statement in this section (4.2) that all mentioned ages hereafter are in calendar years.

p. 10, lines 20-22: Why not show I. norcrossi and S. loeblich in Fig. 3? I. norcrossi could be added to I. helenae (since most other studies referred to have done that). S. loeblichi tend to have fairly confined peaks that may aid in stratigraphic correlation (see comment below on Younger Dryas). Also, I see in the species list in the appendix that there are many Elphidium species (apart from E. clavatum) – if added together they may point out events of lower salinity/freshwater input (see Polyak et al., 2002). I would also like to see the % agglutinated species. Rarer species could be plotted in an appendix figure or supplement.

p. 10, line 29: "(dominant) species generally follow the pattern of the total benthic foraminifera flux". Yes – evidently – better to show total benthic accumulation rate (AR) (the word 'flux' is for something raining down from above) and remove the accumulation rate for individual species in the figure – it does not really mean anything, because accumulation rates only relate to productivity, while species percentages relate to a

high number of other ecological factors. Show only total AR along the species percentage distribution - also because the values can differ by several orders of magnitude between different species so that they are not really comparable. Finally, the accumulation rates depends on the age model, which may not be accurate...(and please plot the total AR as a line).

p. 14, lines 32-36: I believe this has been written before concerning a record from Isfjorden in a publication by Rasmussen et al. 2012? 2013? with the same references. Minimum refer to this publication.

Section 5.2 Younger Dryas: The main question here is: how much of this interval belongs to the Younger Drvas stadial - or is it the Younger Drvas at all? This study has used a standard modern reservoir age correction deltaR of 98+/-37 years. However, it is well known that reservoir ages have varied through time - and the Younger Dryas interval is one of the best investigated periods for changes in reservoir age. DeltaR ages of 300 years have been recorded off UK (several Austin et al. refs), 800-1000 years for the central Atlantic (Waelbroeck et al., 2001), 150-200 years in coastal western Norway (Bondevik et al., 2006) - see also Butzin et al., 2005. More importantly, Hanslik et al., 2010 performed an exercise experimenting with different reservoir age scenarios for the Younger Dryas interval in a record from the Arctic Ocean and settled for 1000 years. With a higher deltaR the date calibrated to 11.9 ka is probably of Holocene age, the age of 12.6 ka may also belong to the Holocene – and the age of 13.2 ka may actually fall within the YD interval. If so, the rise in N. labradorica will occur in the Holocene, which makes more sense, because it signifies retreat of the polar front and less sea ice cover. The following decrease in percent of this species could then be the Pre-Boral Oscillation. Your data just below the suggested YD-bar indicate low productivity and a maximum in sea-ice cover - could this be YD instead? What other benthic foraminiferal speices are present? I pointed out in the beginning that the discussion is very local and only compares and discusses shelf records around Svalbard. This is partly acceptable for the Bølling-Allerød and Holocene intervals, which have been discussed and compared to Nordic seas and Atlantic records on many occasions (but do take a look at Wollenburg papers from the Arctic Ocean). But for the YD, I will request an in-depth comparison to 'outside records' and discussion – records from the Arctic Ocean, Nordic seas and Atlantic Ocean. Further south the YD is easily defined by presence of tephra or other means (see e.g. Austin and Hibbert, 2012, where in addition to tephra, there are 0% N. pachyderma before and after the YD, and 100% in the YD interval). I especially request that the Hanslik-paper and some of Wollenburgs papers are discussed and compared to. This means that the Younger Dryas in core GeoB10817-4 should first and foremost be defined and identified with certainty (as much as possible). This can be done by looking for patterns in the distribution of benthic foraminifera (this is why I asked above to plot more species as they may better indicate the location of YD), but also concentration and accumulation rate data, IRD and grain sizes, isotopes etc. I believe a detailed and thorough discussion of the Younger Dryas interval can lift this manus above being merely a local study comparing mainly with other local studies.

Given the above consideration about reservoir age changes – Figure 7 show some records from the GeoB10817-4 core plotted along the NGRIP ice core. Since your record is not corrected for true reservoir age changes the apparent synchroneity with the ice core is false – marine calibrated 14C ages and ice core years from layer counting are not expected to match (except maybe for parts of the Holocene and depending on the marine core location). As HS1 is known for the same reservoir age problems as the YD (actually even worse, see refs above) the correlation to the ice core here may also be obsolete.

**Minor points:**

p. 4, line 18: 'West' Spitsbergen Current p. 5, line 28: 'Additionally,' p. 6, line 34 and line 35: "at" change to 'to' p. 7, line 22: 'ice volume effect' p. 9, line 36: missing 'the' after "abundant" p. 10, line 28: "flux rate" – flux is a rate – use accumulation rate for benthic foraminifera p. 10, line 32: 'ice volume effect' p. 10, line 33: "lighter" – use the term 'lower' – values cannot be 'light' (or 'depleted' (e.g., p. 14, line 36)) p. 11,

line 36: add 'interstadial' after Bølling p. 13, line 16: 'shelf' p. 14, line 11: missing 'the' after "from" From p. 14 onwards: some awkward wordings here: "synchronously,", "benefitted", "unveil", "a fading", "risen", "go along", "diminished down", please edit. p. 16, line 19: 'slope', not "shelf edge" p. 17, line 37: delete "the" after "into" p. 18, line 23: "on" – I think should be 'onwards'

---

## Referee Comment (RC2) · Anonymous Referee #2 · 7 Jul 2017

This paper looks at the evolution of the Woodfjorden system over the last ∼15,000 years. This system was chosen because of how it interacts (and is impacted) by the warm Atlantic Water that enters the Arctic near Svalbard through Fram Strait. The authors carry out a detailed reconstruction of marine environmental conditions in the area around the fjord to look at temporal evolution in the region and the role of AW (vs glacier dynamics) in driving the observed changes.

This is a well written paper, easy to read and follow. The data is clearly presented, and the conclusions drawn from it make sense. The figure quality is good, and the figures provided are appropriate. So it will definitely make a good and interesting contribution to the literature. That said, there are some small ways that the manuscript could be improved. Most importantly is adding to and strengthening the discussion of Atlantic

[Figure]

Water in the region, as well as something on the processes that could allow it to enter fjord system (like Woodfjorden in this work). As well, at times I find the authors use older references when there are newer papers that would be better to refer to. Thus I would recommend minor revisions.

Given the importance of Atlantic Water to the manuscript, I find section 2 a bit short and incomplete. How much Atlantic Water enters the region, and how does it vary? What drives that variability, i.e. the North Atlantic Oscillation? What depth is it found at today and what sets it? What drives there to be more or less Atlantic Water transport through Fram Strait vs the Barents Sea Opening? How can it be exchanged from the open ocean across a fjord sill (at the very least, there is some discussion of this for Greenland, if not Svalbard)? All of these items are important for the reader to understand so that the authors can link what happens today to their paleoscenarios. The references in section can be brought more up to date as well.

Also, the introduction has old references. I.e. for Fram export, there are newer works then Serreze et al., 2006, for example.

Section 4.2: How would the results change if that one radiocarbon age had not been excluded?

Section 5.1: What does the topography of the fjord system look like with the sea level drop at this time? What does the oceanic Atlantic Water circulation potentially look like at this time?

Section 5.2: Not sure about the forced to submerge comment. If the Atlantic Water is cooled in winter, it will have its density increase and it will sink, irregardless.

Also, if the Ekman transport is northward, that is away from the fjord, so am not sure how that allows the Atlantic Water to flood the fjord.

Figure A1: What are the units of the vertical axis? Shouldn't they be m or dbar? Also, I'd like to see this figure as a panel in the actual paper, not just supplementary material.

Having an understanding of the vertical structure and where the Atlantic Water sits would be useful to the reader.

Finally, it would be good in the discussion section to try to take the results, which are focussed locally on the Svalbard area, and speculate on the wider significance.
* * *

---

## Author Response (AR1)

**Reply to reviewers**

First of all, we would like to thank the editor and the reviewers for the comments that will certainly improve our manuscript. In the following, comments of the reviews are cited in italics followed by our replies (references to pages and lines refer to the revised version without mark-ups).

5 **Reviewer #1**

*The present ecology of foraminifera is very sporadically discussed or described and with only few references. Actually, studies such as by Hald and Korsun, 1997 and Korsun and Hald, 2000 of the distribution of living benthic foraminifera in Svalbard fjords have characterized species in terms of 'ice-proximal', 'ice-distal' (which are useful terms for the present study from a fjord affected by glaciers) and also related them to influx of AW and meltwater*
10 *to the fjords.*

We agree that it might be necessary for the understanding of our interpretation to describe the ecology of foraminifera in more detail (e.g., p. 13, line 29–34). We improved this in the revised version of the manuscript. However, the categorization into "ice-proximal" and "ice-distal" seems to be too simplistic as the occurrence of, e.g., *C. reniforme* and *E. clavatum* (in older publications referred as *E. excavatum*
15 forma *clavata* or grouped as *E. excavatum*) is not restricted to ice-proximal habitat and not even to (Arctic) fjord settings as described in various studies on recent faunas (e.g., Feyling-Hanssen, 1972; Sejrup et al., 2004). Korsun and Hald (1998) explicitly pointed out that the latter two species should not alone be used as proxies for ice-proximal environments. While we cited Hald and Korsun (1997) quite frequently, we agree that it would be useful to refer also to Korsun and Hald (2000) as they describe modern benthic
20 faunas in a fjord from central Spitsbergen. We now included this reference in our manuscript.

*There are references to foraminiferal studies from Iceland – I would think living foraminifera in Svalbard fjords today and from East Greenland and the Arctic Ocean would be more relevant (e.g. Ahrens et al., 1997; Newton and Rowe, 1995; Jennings refs from the Greenland margin; Wollenburg refs from north of Svalbard).*
25 Following the advice of the reviewer, we now added more references to modern faunas of fjords in Svalbard and (east) Greenland (p. 12, line 13; p. 13, line 31–32; p. 16, line 19; p. 18, line 30, 38). We still also refer to other modern shelf faunas, e.g., from Iceland, Novaya Zemlya, the Barents and Kara Sea, because here the benthic foraminiferal faunas are more comparable to our shelf fauna than, e.g., faunas from the Arctic Ocean.

*Often the ecology of the foraminifera is presented as cf. (ref) (e.g. p.14, line 18, p. 15, line 10, p. 17, line 12). I would like to know what these references contain – not having to read the whole publication to find out what is meant here. There is in general too many 'cf.'s' in this manus – one or two is acceptable, but not more. It has to be written out clearly what exactly in all these published papers that is referred to here.*
35 We reduced the number of "cf.´s" significantly (partly these have been used in a wrong way) and give now more details about the ecological preferences.

*'Abstract': line 6: "spanning the last c. 15,000 years". Your record has a six thousand-year gap - this should be made clear from the start. Also, please mention the core name, length and water depth in the abstract.*
40 We added this information (p. 2, line 14–15).

*'Introduction': p. 4, lines 8-11. This is a bit of over statement: the records you refer to are also multiproxy comprising planktic and benthic stable isotopes and planktic and benthic foraminiferal faunas, diatoms, various grain sizes and IRD and are most often also of much higher resolution than your record. Better to state that your*
45 *record is in a more ice proximal setting than the others, and that you can add low-resolution biomarkers (with one or two data points per 1000 years) and a different method for IRD-analysis in fairly high resolution.*

We agree and changed the text - thank you for the suggestion (p. 4, line 11–14).

*p. 5, line 11: give the time interval range between the 5 cm samples and the biomarker samples (also distance between the latter) and the range of time within the 1-cm samples.*

Because the time interval is dependent on the (strongly variable) sedimentation rate, a uniform time interval between the 5cm-steps cannot be given here (ranging from 8 to 123 years per centimetre). Biomarker samples were taken at selected points (e.g., remarkable changes in the other data) but neither at uniform depth intervals nor at uniform time intervals. We used syringe-samples of 1 cm diameter. Accordingly, the analysed sediment within this centimetre is mixed and no time range can be given.

*Chapter 3.1 'Computer tomography': must be written clearer: what is the resolution of counts of the IRD > c. 1 mm and the volume %? I am also unfamiliar with this method and suggest that a simple count of IRD is performed and shown. This would not be a great effort as there are sieved residues from foraminiferal samples and I suggest IRD > 1mm and also 0.5-1 mm be counted for comparison with the computer analyses and for the sake of comparison with published IRD counts.*

The CT-based individual IRD counts and volume percentage calculations have a resolution of 0.35 mm (see p. 5, line 35), while the presented counts were calculated per cm3. Accumulation rates of counted mineral grains (>500 µm) data correspond to the IRD data derived from the CT analyses, but the precision of the CT analyses is much higher as they refer to the entire width of the sediment core and not only to a small sample that has been sieved. Especially the combination of volume percentages and counts per cm3 provides a more realistic "picture" of the IRD input because both data sets have certain disadvantages (see p. 10, lines 18–20). Of course, we could show the accumulation rate of counted mineral grains >500 µm in the appendix (see Fig. A3) as they are already processed but this does not add any new information to the story.

*p. 6. 'Grain size measurements'. Why are sortable silt data not presented and discussed? The manuscript mainly discusses the silt size fraction in terms of ice rafting from sea ice and/or supplied from glaciers. This size fraction can come from glaciers and sea ice yes – but can be current sorted as well. To me the coarsening from the Holocene onwards is because of current sorting – bottom currents are only mentioned on p. 17, line 4. I cannot see why presence of coarser silt should indicate sea-ice rafting, when definitely there is bottom current activity as indicated by certain benthic foraminiferal species. This may also explain the low sea-ice biomarker concentration mentioned on p. 16, lines 32-35. The maxima in 20 my could be more due to current sorting, than ice rafting in my opinion.*

In an Arctic fjord mouth setting the grain size distribution of the sediments can be affected by downslope sediment transport, by sea-ice/iceberg input, by hemipelagic sedimentation and by the action of bottom currents. Hence, the sortable silt approach is difficult to apply, as the various impact factors are hard to differentiate – in contrast to contourite deposits, which are the prime candidates to apply sortable silt. Hass (2002) proposed a method to calculate "IRD-corrected" sortable silt for areas influenced by IRD but his study is based on a sediment core from >1000 m water depth and is, thus, not applicable to our much shallower, near-coastal core site. Nevertheless, it is possible that the coarsening in our record during the Holocene has been caused by increasing current speeds instead of winter sea ice, and, thus, this point has been taken up as an additional possible interpretation (p. 18, line 6–7). *C. lobatulus* (which is commonly associated with high current speeds) shows only low percentages over the entire record and should therefore be interpreted cautiously.

*p. 6, lines 36-37: "several specimens show signs of dissolution or transport" – any particular interval or throughout the record? "Small fragments were not counted since these specimens might be allochtonous" – still, it would be good to see the % fragmentation data and how they compare to species distribution patterns*

As the three dominating species (*N. labradorica*, *C. reniforme* and *E. clavatum*) in our as well as in other high-latitude fjord studies have comparable thick shell walls, which are usually well preserved even when dissolution is recognisable, we did not calculate any "fragmentation index" (as suggested by, e.g., Berger et al., 1982; Pfuhl and Shackleton, 2004; Zamelczyk et al., 2013). Thus, these data do not exist. However, thin-walled foraminifera, e.g. *Globobulimina* spp., are only of very minor relevance for our study area.

*p. 7, line 20: "the endobenthic foraminifera Nonionellina labradorica" – there is no documentation at all in this manuscript about the life mode of N. labradorica – is it shallow infaunal or deep infaunal? What kind of food does it require? Fresh or partly degraded, refractory? Or?*

Unfortunately, this necessary information got lost while editing the manuscript before submission. And as *N. labradorica* is one of our "key species" it is indeed very important information. *N. labradorica* prefers fresh organic matter and is even able to sequester chloroplasts (Bernhard and Bowser, 1999; Cedhagen, 1991). This species shows maximum abundances between one and two centimetres sediment depth (Alve and Bernhard, 1995; Loubere and Rayray, 2016). We implemented this information in the discussion (p. 13, line 29–34).

*Any vital effects on 13C?*

Due to a very complex relationship between foraminiferal $\delta^{13}C$ and $\delta^{13}C_{DIC}$ and a variety of different corrections for a "vital effect" suggested by various authors (e.g., Grossman, 1987; Hesse et al., 2014; Ivanova et al., 2008; Zajączkowski et al., 2010), we did not correct the $\delta^{13}C$ data for this offset. This is also the common approach followed in other studies from the Svalbard area (Klitgaard Kristensen et al., 2013; Rasmussen et al., 2012). As we mainly discuss the offset between $\delta^{13}C$ values of *C. lobatulus* and *N. labradorica*, a correction would not change the general trend/pattern but only the magnitude.

*Later the differences in 13C values between C. lobatulus (epibenthic, ref.?) and N. labradorica are used as an indication of export productivity. You have a low-resolution brassicasterol flux record (that do not match the benthic foraminiferal AR) – how does your 'export production delta13C' record compare to the brassicasterol flux and benthic AR? (they are not shown in the same figure). N. labradorica is a deep infaunal species, and changes in 13C may well be that the species moves up or down in the sediment in search for food, more than a direct signal of the amount of food.*

Similar to other Cibicidinae species like *C. wuellerstorfi*, *C. lobatulus* is a "clinging" epibenthic species that is often found attached to hard substrate like pebbles, molluscs etc. (Dubicka et al., 2015; Linke and Lutze, 1993). The offset between $\delta^{13}C$ ratios of *C. lobatulus* and *N. labradorica* has just recently been approved as a very valuable (export) productivity proxy (Mackensen et al., 2017). These authors also showed that changes in the average living depth (ALD) have only minor influence on the $\delta^{13}C$ values of a certain species probably because species usually calcify in a specific depth or foraminiferal $\delta^{13}C$ ratios reflect average values for species moving up and down in the sediment.

Indeed, there is no point-to-point correlation of $\Delta\delta^{13}C$ and the accumulation rates of brassicasterol and benthic foraminifera. However, as all three data sets are only "proxies" for productivity also influenced by various further factors, they should only be interpreted qualitatively.

*p. 9, line 18. I believe is it not necessary to write 'BP' or 'cal.' I suggest to make a statement in this section (4.2) that all mentioned ages hereafter are in calendar years.*

We followed this suggestion and we removed the "BP" in the text and in all figures (x-axes) after the statement on p. 10, line 1–2.

*p. 10, lines 20-22: Why not show I. norcrossi and S. loeblich in Fig. 3? I. norcrossi could be added to I. helenae (since most other studies referred to have done that). S. loeblichi tend to have fairly confined peaks that may aid in stratigraphic correlation (see comment below on Younger Dryas). Also, I see in the species list in the appendix that there are many Elphidium species (apart from E. clavatum) – if added together they may point out events of lower salinity/freshwater input (see Polyak et al., 2002). I would also like to see the % agglutinated species. Rarer species could be plotted in an appendix figure or supplement.*

*I. norcrossi* s.l. are indeed shown in many other studies, but in most studies the latter species is lumped with *I. helenae* while interpreted as *I. helenae* although both species have different environmental preferences. In our study, *I. norcrossi* actually shows a comparable trend as *I. helenae* but with lower percentages and less pronounced peaks. *S. loeblichi* s.l. likewise never exceeded 8 %. As *Elphidium* spp. have partly very diverging habitats we see no reason for lumping them. Agglutinating species seem to be strongly affected by taphonomic processes as their abundance decreases clearly with increasing core depth. For this reason, agglutinants have been excluded from the calculation of percentages. However,

we see the point of providing the available information and, thus, we added an appendix figure showing rare species including agglutinants (Fig. A4).

*p. 10, line 29: "(dominant) species generally follow the pattern of the total benthic foraminifera flux". Yes – evidently – better to show total benthic accumulation rate (AR) (the word 'flux' is for something raining down from above) and remove the accumulation rate for individual species in the figure – it does not really mean anything, because accumulation rates only relate to productivity, while species percentages relate to a high number of other ecological factors. Show only total AR along the species percentage distribution - also because the values can differ by several orders of magnitude between different species so that they are not really comparable. Finally, the accumulation rates depends on the age model, which may not be accurate (and please plot the total AR as a line).*

We followed this suggestion (see Fig. 5a).

*p. 14, lines 32-36: I believe this has been written before concerning a record from Isfjorden in a publication by Rasmussen et al. 2012? 2013? with the same references. Minimum refer to this publication.*

We now also refer to Rasmussen et al. (2012) (p. 16, line 6).

*Section 5.2 Younger Dryas: The main question here is: how much of this interval belongs to the Younger Dryas stadial – or is it the Younger Dryas at all? This study has used a standard modern reservoir age correction deltaR of 98+/-37 years. However, it is well known that reservoir ages have varied through time – and the Younger Dryas interval is one of the best investigated periods for changes in reservoir age. DeltaR ages of 300 years have been recorded off UK (several Austin et al. refs), 800-1000 years for the central Atlantic (Waelbroeck et al., 2001), 150-200 years in coastal western Norway (Bondevik et al., 2006) – see also Butzin et al., 2005. More importantly, Hanslik et al., 2010 performed an exercise experimenting with different reservoir age scenarios for the Younger Dryas interval in a record from the Arctic Ocean and settled for 1000 years. With a higher deltaR the date calibrated to 11.9 ka is probably of Holocene age, the age of 12.6 ka may also belong to the Holocene – and the age of 13.2 ka may actually fall within the YD interval. If so, the rise in N. labradorica will occur in the Holocene, which makes more sense, because it signifies retreat of the polar front and less sea ice cover. The following decrease in percent of this species could then be the Pre-Boral Oscillation. Your data just below the suggested YD-bar indicate low productivity and a maximum in sea-ice cover – could this be YD instead? What other benthic foraminiferal speices are present?*

This comment points to a general problem we face when developing radiocarbon-based age models for sediment cores. Of course, assuming a stable reservoir age for millennia or even tens of millennia is at best an approximation and changes of the reservoir age through time are very likely. Now, our problem as a community is, that we do have the relevant information only for selected time intervals in specific regions – what means we lack information for most regions and time intervals (see Ślubowska-Woldengen et al. 2007 for the Svalbard region). And when looking to the Younger Dryas (YD) record from our shallow study area at the northern Svalbard shelf, reported reservoir ages for the Arctic Ocean of ~1000 years are probably less applicable as values of 150–200 years from comparable shallow settings off Norway. But, at the end we do not know. Also the comparison of the records of individual species would be somehow "floating" as also for other regional records, no better stratigraphy is available. And using a specific species to maybe identify the YD sediments, bears the risk to put our "idea" about the response of benthic foraminifera to the YD event to the base of our stratigraphic interpretation. Independent indicators to localise the YD, such as the Vedde ash, unfortunately are not applicable in these high northern latitudes (although Zamelczyk et al., 2012, reported the finding of the Vedde ash in a sediment core from the Fram Strait at 78°N thereby, however, admitting that a redistribution by bottom currents or sea ice cannot be excluded).

Thus, the main point to make here is that we cannot guarantee that what we identified as the YD in our record indeed reflects the YD (on this level of stratigraphic resolution this counts for most sedimentary records) as reservoir ages of 100, 200 or 1000 years for this period is likewise probably. Therefore, we followed the most often applied approach for sedimentary records from the Svalbard region and continued to work with stable reservoir ages – due to the lack of any information on "paleo-reservoir ages". This allows easy comparison of all these studies by now and a simple and common "re-dating" approach once

new information on past reservoir ages becomes available. But, of course, we also see the point of the reviewer about the difficulties in interpreting "our YD record" and we added a statement to the text outlining the stratigraphic uncertainties resulting from unknown reservoir age histories (p. 6, line 18–24).

5 *I pointed out in the beginning that the discussion is very local and only compares and discusses shelf records around Svalbard. This is partly acceptable for the Bølling-Allerød and Holocene intervals, which have been discussed and compared to Nordic seas and Atlantic records on many occasions (but do take a look at Wollenburg papers from the Arctic Ocean). But for the YD, I will request an in-depth comparison to 'outside records' and discussion – records from the Arctic Ocean, Nordic seas and Atlantic Ocean. Further south the YD is easily defined*

10 *by presence of tephra or other means (see e.g. Austin and Hibbert, 2012, where in addition to tephra, there are 0% N. pachyderma before and after the YD, and 100% in the YD interval). I especially request that the Hanslik-paper and some of Wollenburgs papers are discussed and compared to.*

We already referred to "outside records" from the North Atlantic and now we added some more comparisons (p. 14, line 38–39; p. 15, line 2–7; p. 15, line 32–37). Unfortunately, we cannot use planktic

15 foraminifera (or their absence) as tracer for the YD because they are extremely rare in our entire record due to the shallow water depth and the littoral position of our core site. Seeing the stratigraphic uncertainties discussed above, we do not see any chance to add an in-depth comparison to 'outside records' for the YD.

20 *This means that the Younger Dryas in core GeoB10817-4 should first and foremost be defined and identified with certainty (as much as possible). This can be done by looking for patterns in the distribution of benthic foraminifera (this is why I asked above to plot more species as they may better indicate the location of YD), but also concentration and accumulation rate data, IRD and grain sizes, isotopes etc. I believe a detailed and thorough discussion of the Younger Dryas interval can lift this manus above being merely a local study comparing mainly*

25 *with other local studies.*

As already mentioned above, we do not see any chance to follow this suggestion. All available records from shelf settings around Svalbard have the same problem in identifying e.g., the YD. Thus, detailed benthic foraminifera distribution patterns might be used for high-resolution correlation of records but do not help in assessing their real age. The only way would be to assume a certain environmental, climatic

30 and hydrographic setting for the YD and to estimate the foraminiferal response to those – what hardly would be convincing. Again, in most studies that we refer to (and compare our data to) a linear regional reservoir age is used to develop the age model. For comparison with those studies it is more useful to follow the same approach unless more precise age models can be developed.

35 *Given the above consideration about reservoir age changes – Figure 7 show some records from the GeoB10817-4 core plotted along the NGRIP ice core. Since your record is not corrected for true reservoir age changes the apparent synchroneity with the ice core is false – marine calibrated 14C ages and ice core years from layer counting are not expected to match (except maybe for parts of the Holocene and depending on the marine core location). As HS1 is known for the same reservoir age problems as the YD (actually even worse, see refs above)*

40 *the correlation to the ice core here may also be obsolete.*

As we do not know the exact "true" reservoir age, whether during the YD nor during the HS1, our correlation is probably false following the line of thought of the reviewer – but it could be right … Basically, we feel that we developed the age model according to the state-of-the-art, knowing that is far from being perfect, and, within the uncertainties, we feel we also can use it to compare our record with

45 ice core records as it is done in many studies.

*Minor points:*
*p. 4, line 18: 'West' Spitsbergen Current p. 5, line 28: 'Additionally,' p. 6, line 34 and line 35: "at" change to 'to'*
*p. 7, line 22: 'ice volume effect' p. 9, line 36: missing 'the' after "abundant" p. 10, line 28: "flux rate" – flux is a*

50 *rate – use accumulation rate for benthic foraminifera p. 10, line 32: 'ice volume effect' p. 10, line 33: "lighter" – use the term 'lower' – values cannot be 'light' (or 'depleted' (e.g., p. 14, line 36)) p. 11, line 36: add 'interstadial' after Bølling p. 13, line 16: 'shelf' p. 14, line 11: missing 'the' after "from" From p. 14 onwards: some awkward wordings here: "synchronously,","benefitted", "unveil", "a fading", "risen", "go along", "diminished down",*

*please edit. p. 16, line 19: 'slope', not "shelf edge" p. 17, line 37: delete "the" after "into" p. 18, line 23: "on" – I think should be 'onwards'*

We corrected these mistakes.

**Reviewer #2**

5   *Most importantly is adding to and strengthening the discussion of Atlantic Water in the region, as well as something on the processes that could allow it to enter fjord system (like Woodfjorden in this work). As well, at times I find the authors use older references when there are newer papers that would be better to refer to. Thus I would recommend minor revisions.*

*Given the importance of Atlantic Water to the manuscript, I find section 2 a bit short and incomplete. How much*
10   *Atlantic Water enters the region, and how does it vary? What drives that variability, i.e. the North Atlantic Oscillation? What depth is it found at today and what sets it? What drives there to be more or less Atlantic Water transport through Fram Strait vs the Barents Sea Opening? How can it be exchanged from the open ocean across a fjord sill (at the very least, there is some discussion of this for Greenland, if not Svalbard)? All of these items are important for the reader to understand so that the authors can link what happens today to their paleoscenarios.*
15   *The references in section can be brought more up to date as well. Also, the introduction has old references. I.e. for Fram export, there are newer works then Serreze et al., 2006, for example.*

The amount of Atlantic Water entering the Arctic Ocean/the Svalbard area is depending on variations of the NAO and the AO (p. 4, line 21–25). Additionally, large seasonal differences are evident in the modern Atlantic Water circulation (p. 5, line 2–5). Various mechanisms allow Atlantic Water to enter the fjord,
20   including barotropic instabilities and wind-induced upwelling (p. 5, line 8–12). We included that information about the modern oceanography and we also included some more recent studies (Beszczynska-Möller et al., 2012; von Appen et al., 2016). Nevertheless, we also stick to the "older" references from the 1980s and 1990s as these are the "pioneering" studies regarding the circulation patterns in the Svalbard area.

*Section 4.2: How would the results change if that one radiocarbon age had not been excluded?*

As the excluded radiocarbon age is a "reversal" a linear interpolated age–depth model including this age is not possible. We tried to implement that age in an age–depth model developed with the Bayesian software BACON (Blaauw and Christen, 2011), but the model yielded no reasonable results.

*Section 5.1: What does the topography of the fjord system look like with the sea level drop at this time? What does the oceanic Atlantic Water circulation potentially look like at this time?*

The Woodfjord has no sill in the narrower sense, but on the seaward side of our core, a shallower ridge connects the peninsula Reinsdyrflya and the island Moffen (see Fig. 1c). Thus, a sea level drop might
35   have slightly reduced the Atlantic Water inflow, if at all. More important might have been the fact that the inflow of Atlantic Water was not restricted by the "competing" Coastal Current, because this current (the northern extension of the East Spitsbergen Current) develops not before the deglaciation of the Barents Sea. We added this point to the discussion (p. 13, line 18–20, line 38–39; p. 14, line 1–3).

40   *Section 5.2: Not sure about the forced to submerge comment. If the Atlantic Water is cooled in winter, it will have its density increase and it will sink, irregardless.*

We assume that the intensive meltwater discharge during the deglaciation led to a thick surface water layer, thus, Atlantic Water advection was only possible in subsurface depths, partly even reaching the sea floor (see p. 13, line 1–3). During the Younger Dryas, the meltwater outflow may have been slightly
45   reduced enabling Atlantic Water to enter shallower water masses, e.g., intermediate or even subsurface waters. Additionally, the benthic foraminiferal fauna rather reflects the spring/summer situation than winter conditions. Thus, winter mixing might not be indicated by the fauna.

*Also, if the Ekman transport is northward, that is away from the fjord, so am not sure how that allows the Atlantic*
50   *Water to flood the fjord.*

Wind-induced Ekman transport would affect surface waters. If surface waters are deflected to the north, underlying intermediate waters (in this case Atlantic Water) would be upwelled to the south and were, therefore, able to enter the shelf and the fjord, respectively (See also Fig. 9b). We added these details regarding Ekman transport (p. 14, line 9–10).

*Figure A1: What are the units of the vertical axis? Shouldn't they be m or dbar? Also, I'd like to see this figure as a panel in the actual paper, not just supplementary material. Having an understanding of the vertical structure and where the Atlantic Water sits would be useful to the reader.*

The units of the y-axis are meter below sea level (mbsl). We mentioned the units in the figure caption and moved the figure into the main text (now Fig. 2).

*Finally, it would be good in the discussion section to try to take the results, which are focussed locally on the Svalbard area, and speculate on the wider significance.*

We compared our data with studies beyond the Svalbard region (as mentioned above) in the revised version (e.g., comparisons to studies from Greenland, Norway, Germany, Scottish Highlands and Newfoundland; p. 14, line 38–39; p. 15, line 2–7; p. 15, line 32–37). Additionally, we included the study of Łącka et al. (2015) from the Storfjord Trough (southern Svalbard) in the discussion (p. 13, line 11–12; p. 15, line 4–5; p. 18, line 13–15) as suggested by those authors (see short comment; doi:10.5194/cp-2017-53-SC1, 2017).

[revised manuscript text omitted]

(b) depth [cm]

DISTURBED SECTION

IRD
bioturbation
disturbed section

(c) cal. age [ka BP]

DISTURBED SECTION

depth [cm]

sedimentation rate [cm ka⁻¹]

(a) depth [cm]

clast-size [mm]
64  0.5

[Figure]

**Fig. 23: a)** CT scan of the core GeoB10817-4. Left to right: orthogonal profile; interpreted image (clasts and bioturbation, s. legend); clast-size distribution (0–20 vol. % of clasts: blue to red, respectively). Yellow rectangle marks the disturbed section also shown in (b). **b)** CT images (orthogonal and interpreted) of the disturbed section (yellow rectangle; definition of boundaries: see sect. 4.1 and  Fig. A2 in the appendix). **c)** Age–depth plot for sediment

core GeoB10817-4: Diamonds show calibrated (median) ages with error bars (Table 1); solid black line shows linear interpolation excluding an outlier at 257 cm. Blue solid line shows corresponding sedimentation rates.

[Figure]

**Fig. 4: a)** Grain-size distribution [vol. %] of fine siliciclastic sediments (0–63 μm) derived from laser diffraction particle size analyses. Green rectangles: selected grain-size distributions as shown in (d). **b)** mean grain size [μm] of siliciclastic sediments. **c)** Ice rafted debris (IRD): volume percentage (solid red line) and clasts per cm$^3$ (reddish shading) derived from CT analysis. Stars mark peaks in vol. % corresponding to single large clasts. Grey vertical shadings indicate cold periods: Preboreal Oscillation (PBO), Younger Dryas (YD) and Heinrich Stadial 1 (HS1). Dark grey diamonds: calibrated radiocarbon dated depths with error ranges. **d)** selected grain-size distributions [vol. %] at ~8.8 ka (light green line), ~11 ka (dark green line) and ~13.9 ka  (green line).

[Figure]

[Figure]

**Fig. 45: a) Accumulation rates of benthic foraminifera (ARBF) and (b–h) rRelative percentages (lines) and accumulation rates (shadings) of dominant calcareous benthic foraminifera. ab) *Cibicides lobatulus*. bc) *Buccella frigida* s.l. ed) *Islandiella helenae*. de) *Nonionellina labradorica*. ef) *Cassidulina neoteretis*. fg) *Elphidium clavatum*. gh) *Cassidulina reniforme*. Grey vertical shadings: s. Fig. 3 
[revised manuscript text omitted]